# The topological requirements for robust perfect adaptation in networks of any size

Robyn P. Araujo [1,2] & Lance A. Liotta[3]

Robustness, and the ability to function and thrive amid changing and unfavorable environments, is a fundamental requirement for living systems. Until now it has been an open question how large and complex biological networks can exhibit robust behaviors, such as perfect adaptation to a variable stimulus, since complexity is generally associated with fragility. Here we report that all networks that exhibit robust perfect adaptation (RPA) to a persistent change in stimulus are decomposable into well-defined modules, of which there exist two distinct classes. These two modular classes represent a topological basis for all RPA-capable networks, and generate the full set of topological realizations of the internal model principle for RPA in complex, self-organizing, evolvable bionetworks. This unexpected result supports the notion that evolutionary processes are empowered by simple and scalable modular design principles that promote robust performance no matter how large or complex the underlying networks become.

[1] School of Mathematical Sciences, Queensland University of Technology, Brisbane, QLD 4000, Australia. [2] Institute of Health and Biomedical Innovation (IHBI), 60 Musk Avenue, Kelvin Grove, Brisbane, QLD 4059, Australia. [3] Center for Applied Proteomics and Molecular Medicine, George Mason University, 10920 George Mason Circle, Manassas, Virginia 20110, USA. Correspondence and requests for materials should be addressed to R.P.A. (email: r.araujo@qut.edu.au)

Robust perfect adaptation (RPA) is the ability of a system to generate an output that returns to a fixed reference level (its 'set point') following a persistent change in input stimulus, with no need for tuning of system parameters[1–3]. RPA has been widely observed throughout biology[1,4–11], at the cellular level (signal transduction, gene regulation, protein interaction networks[6–8]), in sensory systems[6,10,11], at the whole-organism level in mammals[9], and during development[12,13]. For example, mammalian plasma calcium concentration exhibits perfect adaptation to persistent changes in calcium export (e.g., lactation), or influx (e.g., diet changes or bone remodeling), thereby keeping plasma calcium within very tight tolerances as calcium demands vary[9]. In addition, perfect adaptation enables a biological system to reset itself following a perturbation, in order to maintain responsiveness to subsequent variations in external stimuli[3]. The RPA property thus promotes high sensor sensitivity, while increasing the dynamic range, regardless of the intensity, or the variations, in the average stimulus[1–3,14,15].

Importantly, while RPA confers many benefits to living systems, loss of the RPA property in networks that require it could lead to disease (e.g., ras-mediated oncogenesis[3,16]), reduced fitness[1], or death[17]. During development, and throughout evolution[1], biologic networks—or bionetworks—grow to enormous size and molecular complexity, apparently without any compromise in robustness. Why isn't this growth in complexity associated with heightened fragility, or instability, or a loss of requisite function[18–20]?

In this connection, it is essential to recognize that biological systems differ in fundamental ways from engineering control systems. Molecular signaling networks are self-organizing, self-regulating, adaptable, and evolvable, and as such, are comprised of elements that must serve both as the transmitted signals and their own controllers. Unlike their designed counterparts in engineering control systems, bionetworks do not have the luxury of employing specially-designed, dedicated components whose purpose is to sense or control biochemical signals. Although asymptotic tracking problems (of which RPA to constant exogenous inputs is a special case) have been studied extensively for engineering systems[21–23], how can we understand the mechanisms governing robust performance in the context of the self-organizing, self-regulating autonomous systems arising in biology, or that become deranged in disease[24–26]?

Until now, three basic approaches have been used to understand the allowed topologies for RPA networks[2,3,5,8,14,18,19], and they have only provided answers for very small systems. The first approach is to model a small, well-defined, and well-studied natural system that exhibits RPA; a classic example of this approach is the work of Barkai and Leibler on RPA in bacterial chemotaxis[5]. A second approach is to build trial modifications of input–output network connection maps, hoping to find circuit designs that have the RPA property[3,18,19]. Both of these approaches consider the RPA problem via ad-hoc modeling. By contast, a third approach is to undertake high-throughput computational searches to comprehensively study very small networks, typically only containing two or three components[2,13]. Ma et al.[2] used this approach to suggest that, for three-node networks, only two types of signaling motif were capable of implementing RPA.

Crucially, such ad-hoc or high-throughput strategies are impractical for most networks in living and evolving systems, which can contain huge numbers of interacting molecules, and for which we have limited or no a priori knowledge of component and pathway interconnections, reaction kinetics, or system parameters[1,17,18]. If computational search strategies could be extended to networks with more than three nodes, would this reveal any RPA network topologies that are distinct from the two

basic network designs already discovered for three-node RPA networks? Do the topological requirements for RPA in large systems increase in complexity, or change qualitatively, along with the growth of the system, or must larger systems simply replicate the same basic design principles used by smaller systems? Could there exist universal topological principles that characterize all RPA-capable network designs[18]?

In the present study we provide definitive answers to these questions. In stark contrast to previous work on the RPA problem (see[3,18] for recent reviews), we develop a topological framework, and a new set of unifying definitions, that is able to interpret and account for the flow and control of biochemical signals through a network. This global and top-down methodology allows us to describe the full set of possible network topologies for achieving RPA in arbitrarily large and complex networks, involving any number of interacting components, with no prior assumptions as to how the components are interconnected, or the kinetics of any reactions. Remarkably, we show that all networks, no matter how large or interconnected, have just two distinct mechanisms at their disposal, corresponding to two distinct types of integral control, in order to implement RPA. Each of these two mechanisms generates a rich class of well-defined network topologies—'modules'—containing previously unrecognized architectural features that are too complex to be observed in three- or four-node networks. Most importantly, we show that these two rich and distinct classes of modules represent a topological basis for the solution to the RPA problem: that is, the full set of all possible RPA-capable networks can be expressed via the interconnections of these special modules, subject to well-defined modular connectivity rules.

## Results

**General schema for identifying RPA topologies**. In order to specify the complete solution space to the general RPA problem, we derive and analyze a suitable algebraic condition that we refer to as the RPA Equation (see Methods). This equation accounts for all possible interactions and interconnections in a network of arbitrary size, and establishes a special case of the Internal Model Principle (IMP) from which topological structures may be deduced. The equation takes the form of a particular Jacobian determinant, which is required to take a zero value for all system inputs, $I$.

At the broadest level, we interpret the (signed) terms of the RPA equation's determinant expansion as a set, $\mathbf{R}$, and recognize that partitions of $\mathbf{R}$ may exist for which the contents of every subset can sum to zero independently of the contents of every other subset. We proceed to identify general conditions under which such 'independently adapting subsets' can exist by accounting for the topological information contained within each term. In particular, we show in Supplementary Note 2 that each member of $\mathbf{R}$ represents a unique product of the three fundamental mathematical elements of signal transmission— (1) a route factor (a pathway through the network, corresponding to an unbroken sequence of node-node interactions commencing at the input node and ending at the output node), (2) circuit products (corresponding to multi-node cycles, i.e., feedback loops), and (3) 'kinetic-multipliers' (single-node cycles, which encode important properties of the reaction kinetics at the node in question).

The cornerstone of our methodology is to identify conditions under which an arbitrary subset of $\mathbf{R}$ can sum to zero for all $I$, thus solving a local adaptation equation (LAQ), with no possible further sub-division into smaller adaptive subsets. We refer to such a subset as a 'minimally adaptive (MA)-subset', of which we identify two distinct types: singleton MA-subsets (hereafter, S-

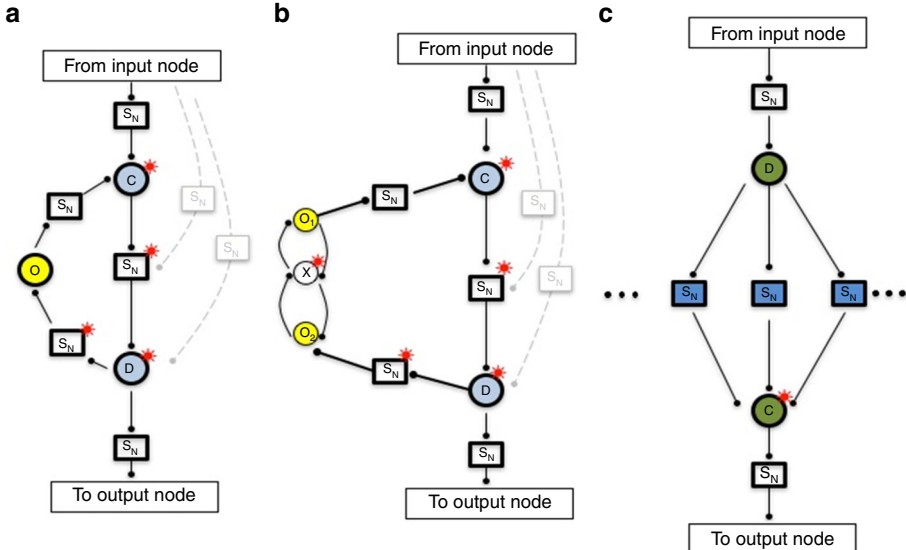

**Fig. 1** Two module classes are a topological basis for RPA networks. The essential architecture of the two RPA basis modules: the Opposer Module (**a**, **b**) and the Balancer Module (**c**). All nodes that exhibit the RPA property are denoted with a superposed red asterisk. The symbol $S_N$ indicates that any motif or sub-network may be fully embedded at the indicated position. Interactions among nodes are indicated with a solid dot, to indicate that the interactions may be either activating or inhibitory in principle. All nodes depicted below may also (optionally) be regulated by extramodular nodes that exhibit the RPA property (blind regulations) due to the activity of other RPA modules in the larger network (not shown, for clarity). These modules rely on special computational nodes—opposer nodes, balancer nodes, and connector nodes—all of which use special classes of reaction kinetics to orchestrate their RPA-generating functions (see Supplementary Note 4). **a** An Opposer Module with a single opposer node (denoted O in yellow). Opposer Modules are characterized by a feedback architecture, defined by an apex node (C) and a base node (D, both in blue). All routes that are contiguous with the route segment C—D are opposed by the opposer node—that is, all routes that feed into the module at some node between C and D, inclusive. The module could therefore include any of optional regulations indicated in feint. **b** An Opposer Module with a two-node opposing set. Structure of the module is as for part (a) except that two opposer nodes now work together in concert within a pair of interlinked circuits embedded into the feedback architecture. Opposer nodes are part of a master set, which in this case comprises {O1, X, O2}. **c** A Balancer Module is characterized by a collection of (at least two) parallel route segments, which diverge upstream at D and reconnect downstream at the connector node, C, both indicated in green. All nodes embedded into those route segments are balancer nodes, indicated in blue. The three dots on either side of the interior $S_N$ elements indicate that the module may contain any arbitrary number of route segments between D and C

sets), whose single term must be able to assume a zero value for all *I*, and multi-term MA-subsets (hereafter, M-sets), all of whose terms must solve their LAQ with strictly non-zero values for all *I*.

S-sets and M-sets are generated (that is, their LAQ is solved) by distinct mechanisms that are orchestrated by special classes of reaction kinetics at one or more key network nodes. For S-sets, we refer to the generating mechanism as 'opposition', and the special class of reaction kinetics that execute the mechanism as 'opposer kinetics'. The 'balancing' mechanism that generates M-sets, on the other hand, requires two different types of reaction kinetics working together in collaboration—'balancer kinetics' and 'connector kinetics'—at distinct nodes. We will see later that these special classes of reaction kinetics play remarkable computational roles in RPA-capable networks.

**S-sets and the creation of Opposer Modules**. S-sets are created by a mechanism we refer to as opposition, since a factor in the cycle component of the term assumes a zero value, thereby opposing that term's route component. The mechanism is transacted by an opposer node, $P_o$, which appears in the form of a kinetic multiplier, and whose reaction kinetics (opposer kinetics) must satisfy $\frac{\partial f_o}{\partial P_o} = 0$ at steady-state, for all *I*.

Now, a node can only exhibit opposer kinetics if it participates in a feedback loop (Theorem 1, Supplementary Note 2), making opposition a circuit-based mechanism. We show that the mathematical requirements of opposer kinetics imply a well-defined class of possible chemical reaction forms, which we describe in Supplementary Note 4. In addition, an opposer node

requires a single independent regulator, which must participate in a common circuit with the opposer, in order to implement these reaction kinetics. This requirement for a single independent regulator implies that an opposer node cannot occur at the junction of two independent feedback loops, for instance.

Importantly, a single opposer node will oppose (assign to S-sets) all instances in **R** of a particular route if and only if (a) it is disjoint from the route, and (b) it participates only in circuits that are contiguous with the route (Theorem 2, Supplementary Note 3). From this it follows that a single opposer node can only partially oppose a route if it participates in even one circuit that is disjoint from the route in question. Nevertheless, all instances of a route could still be assigned to S-sets by a collection of two or more opposer nodes working together in concert. We refer to such a collection of opposer nodes as an 'opposing set'.

We identify a strict set of topological conditions for which a collection of opposer nodes, $\{P_{o1}, \ldots, P_{om}\}$ constitutes an opposing set for a particular route in Theorem 3 of Supplementary Note 3. The topological requirements of opposing sets specified in Theorem 3, combined with the requirement for each opposer node to have a single independent regulator in a common circuit, define a rich class of network topologies associated with the opposition mechanism: a collection of opposer nodes distributed to a set of interlinked circuits, embedded into a feedback loop that is contiguous with the route being (fully) opposed. In this sense, a single opposer (with no disjoint circuits relative to a route it fully opposes, and is embedded alone into a contiguous circuit), may be considered a trivial opposing set—a special case which vacuously satisfies the conditions of Theorem 3.

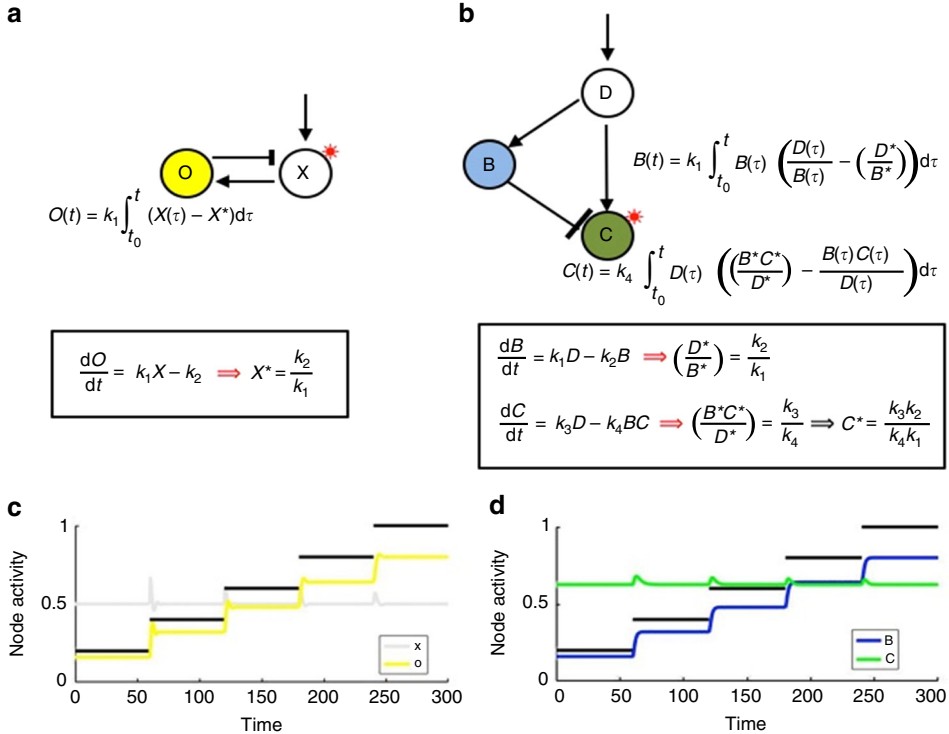

**Fig. 2** Integral control in opposer and Balancer Modules. Solving the RPA problem in general networks relies on the computation of two different types of integral—one for each type of RPA basis module. **a** In an Opposer Module, each opposer node computes a tracking error that involves the activity of its single independent regulator. For the minimal example of an Opposer Module shown here, the single opposer node O exhibits a simple form of opposer kinetics which is zero-order in its substrate, with single independent regulator X. The opposer thereby computes the tracking error between the activity of X and its (input-independent) steady-state value, $X^*$. **b** In a Balancer Module, it is the role of the balancer nodes to compute a steady-state activity ($B^*$) that is a linear function of the steady-state of the diverter node ($B^*$). Each balancer thereby computes an integral that involves the deviation of the nodes' activities from a linear relationship. This upstream computation then creates the conditions for the connector node to track a steady-state value ($C^*$) that is independent of the activity of the D-node through the computation of an integral involving its direct regulators. For the minimal example of a Balancer Module shown here, the balancer tracks the simplest linear relationship with the diverter activity, namely a proportional relationship. **c** Numerical simulations for the simple Opposer model shown in **a**: input ($I$) (shown in black) increases from 0.2 to 1 in steps of 0.2. X-equation: $dX/dt = k_3(I/O) - k_4X$; parameters: $k_1 = k_4 = 1$, $k_2 = 0.5$, $k_3 = 0.4$. **d** Numerical simulations for the simple Balancer model shown in **b**. Input ($I$) (shown in black) increases from 0.2 to 1 in steps of 0.2. D-equation: $dD/dt = k_5I - k_6D$; parameters: $k_2 = k_4 = k_5 = k_6 = 1$, $k_1 = 0.8$, $k_3 = 0.5$

In Fig. 1a, b, we depict the class of network topologies corresponding to the mechanism of an opposing set, illustrating the case of a single opposer node, as well as a simple version of a two-node opposing set. We refer to these network topologies hereafter as Opposer Modules.

**Computational role of Opposer Nodes**. We illustrate in Figs. 2a and 3 (with additional analysis in Supplementary Note 4) that opposer nodes calculate an integral of a tracking error—the difference between some network quantity (e.g., the activity of a particular node) and its steady-state value, the latter being determined purely by system parameters, rather than the magnitude of the system input.

For a single opposer node, the tracking error in question corresponds to the error in the activity of the single independent regulator (Fig. 2a). Since the opposer and its regulator participate in a common circuit, the computation of this integral constrains the regulator node to exhibit the RPA property. In an opposing set, on the other hand, each opposer in the collection computes the integral of a unique tracking error, involving various combinations of nodes in the master set (Fig. 3). All nodes featuring in these various tracking errors exhibit the RPA property due to the combined effect of the multiple integrals; the opposer nodes themselves, by contrast, never exhibit the RPA property (Supplementary Note 4).

It is apparent that these more complex arrangements of opposer nodes are indeed employed in certain gene regulatory circuitries since we show in Supplementary Note 5 that the recently identified phenomenon of antithetical integral feedback[3,14] is actually an instance of a two-node feedback opposing set with single input/output node. Some additional examples of opposing sets are depicted in Fig. 3, with a more general representation of opposing sets given in Fig. 4 (further details in Supplementary Note 3).

**M-sets and the creation of Balancer Modules**. Although the opposition mechanism is orchestrated by feedback loops, along with the collections of opposer nodes embedded into them, the balancing mechanism is route-based. Indeed, we show that an M-set must contain at least two distinct routes (Theorem 4, Supplementary Note 3). A consequence of this requirement is that networks in which a single node acts as both input and output cannot invoke the balancing mechanism to achieve RPA, and must instead rely on opposition (Corollary 2, Supplementary Note 3).

Obtaining a solution to the LAQ places constraints on the reaction kinetics of all nodes within certain route segments, and any fully embedded feedback loops, for the particular routes contained in the proposed M-set. The interconnectivities among these constrained nodes delineate a specific sub-network topology

that implements the balancing mechanism. We refer to this topological configuration hereafter as a Balancer Module, and present a general schematic for the module in Fig. 1c. As shown, a Balancer Module is characterized by a 'diverter node' (D-node) at the apex, and a 'connector node' (C-node) at the base of the

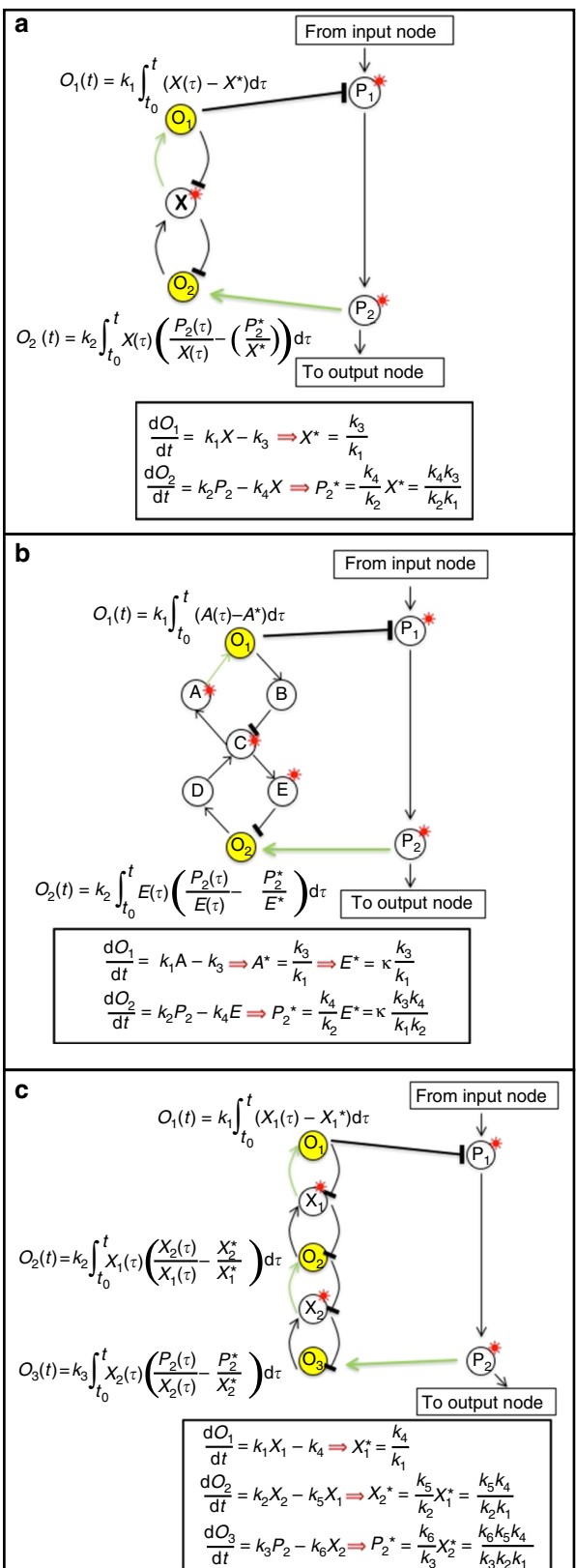

module. Separating the D- and C-nodes are a collaborating set of balancer nodes, comprising all members of the route segments between the D-node and the C-node, for the M-set in question, along with any feedback loops fully embedded into those route segments. The requisite reaction kinetics for balancer nodes constrain their steady-state values to reside on a 'flat manifold' (Theorem 5, Supplementary Note 3)—that is, the steady-state for each balancer is a linear function of the steady-state of the D-node. Full details on the class of reaction forms that implement balancer kinetics are given in Supplementary Note 4; a key topological requirement for the implementation of balancer kinetics is for the set of collaborating balancer nodes to have a single independent regulator—namely, the D-node. The set of balancer nodes must also collaborate with one additional node— the connector node—whose special reaction kinetics (Supplementary Note 4) complete the balancing act for the M-set, allowing its terms to sum to zero for all $I$.

The balancing mechanism thus requires a computational collaboration between two distinct types of nodes: a collection of one or more balancer nodes, along with a single connector node. The reaction kinetics for these two-node types implement a form of integral control that is distinct from the integral feedback control that characterizes the activity of opposer nodes. As we illustrate in Fig. 2b for a very simple Balancer Module, the computational function of the balancer nodes is to linearize their steady-state responses to the activity of the D-node. This upstream linearization creates the conditions that allow the connector node to compute a particular integral—one which constraints the connector to track a steady-state value that is independent of the D-node (and therefore, of the input to the network). In a Balancer Module, then, the connector node alone exhibits the RPA property; the D-node and all balancer nodes, by contrast, assume steady-state values that vary with network input.

**RPA basis sets and their corresponding basis modules.** Armed with an understanding of the two basic RPA-generating

**Fig. 3** Integral feedback control via three kinds of opposing sets. Three different examples of opposing sets, with the nature of the integral computed by each individual opposer node indicated in each case. Opposer nodes are indicated in yellow; a red superposed asterisk indicates nodes within each module that exhibit the RPA property due to the combined activities of the opposing set. Not only do all opposer nodes in the set contribute to an overall computation that allows one or more nodes in the opposed route to exhibit RPA, but one or more nodes in the master set must also exhibit the RPA property as part of this computation. This phenomenon has important implications for the interconnectivity of modules, through the possibilities for either live or blind outgoing regulations from the module. **a** A two-node opposing set $\{O_1, O_2\}$, with associated master set $\{O_1, X, O_2\}$. $O_1$ imparts the RPA property to X, through the computation of a tracking error involving X. With X thus equipped with the RPA property, $O_2$ imparts the RPA property to $P_2$ through the computation of a tracking error involving both X and $P_2$. **b** A two-node opposing set $\{O_1, O_2\}$, with associated master set $\{O_1, A, B, C, D, E, O_2\}$. $O_1$ imparts the RPA property to A, through the computation of a tracking error involving A. The steady-state activities of C and E are dependent on A, so these two nodes also exhibit the RPA property as a consequence. With E thus equipped with the RPA property, $O_2$ then imparts the RPA property to $P_2$ through the computation of a tracking error involving both E and $P_2$. **c** A three-node opposing set $\{O_1, O_2, O_3\}$, with master set $\{O_1, X_1, O_2, X_2, O_3\}$. $O_1$ imparts the RPA property to $X_1$, through the computation of a tracking error involving $X_1$. With $X_1$ thus equipped with the RPA property, $O_2$ imparts the RPA property to $X_2$, through the computation of a tracking error involving $X_2$ and $X_1$. With $X_2$ now equipped with the RPA property, $O_3$ can impart the RPA property to $P_2$ through the computation of a tracking error that involves both $P_2$ and $X_2$

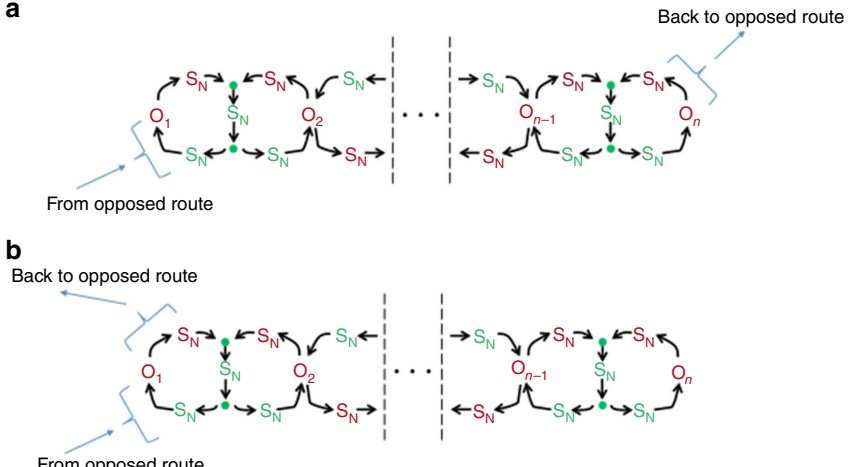

**Fig. 4** The general structure of opposing sets. A general representation of the topology of an opposing set—an arrangement of interlinked circuits that are disjoint from the route they oppose, but embedded in some circuit that is contiguous with the route. We refer to the set of all nodes comprising these interlinked disjoint circuits as the master set associated to the opposing set (see Theorem 3). The requirement for each opposer node to have a single independent regulator requires there to be one, and only one, opposer node in each interlinked circuit. This also requires the opposer node from one circuit to be regulated directly or indirectly by a node in the opposed route; all other opposer nodes in the opposing set will have a single independent regulator that is internal to the collection of disjoint circuits. The opposer nodes $O_i$, and their dependents (all marked in red) do not exhibit RPA. The single independent regulator of each opposer node, and its dependents (all marked in green) do exhibit the RPA property. Thus, any outgoing regulation from a node marked in red will be a live regulation; any outgoing regulation from a node marked in green will be a blind regulation. This has important implications for the interconnections of the module with other parts of the RPA-capable network. The symbol $S_N$ indicates that any motif or sub-network may be fully embedded at the indicated position. **a** For illustrative purposes, an example is shown where the opposer node from one of the disjoint circuits has a single independent regulator that comes from the opposed route; the opposer node, or one of its dependents, from some other disjoint circuit in the collection then regulates the opposed route, thus forming a contiguous circuit into which the opposing set is embedded, as required. **b** For illustrative purposes, an example is shown where a single opposer node in the opposing set is regulated by a route note, and also regulates a route node, thus forming the requisite contiguous circuit for the embedding of the opposing set

mechanisms, we now turn our attention to the central concern of this work—the topological characterization of the set of all possible networks capable of exhibiting RPA.

We note at this point that the balancing mechanism described in the preceding section automatically balances all copies of all routes in **R** that contain the noted route segments between the D- and C- nodes, regardless of whether those terms were selected in the original M-set (see Supplementary Note 5). The union of the M-set with all such terms thus represents the independently adapting subset of **R** associated to the M-set. The Balancer Module depicted in Fig. 1c thus represents the class of network topologies that correspond to an independently adapting (balancing) subset of an RPA equation.

Moreover, any route in a network that is only partially opposed must have copies that are balanced. Since a balancing mechanism will automatically balance all copies of its routes, such a partially opposed route is redundantly opposed. As such, the independently adapting subset associated to any opposition mechanism (via opposing sets) comprises the union of only those terms in **R** whose routes are fully opposed by the mechanism. The union of S-sets generated by partial opposition of a particular route should thus be absorbed into the independently adapting subset associated with the relevant balancing mechanism.

From the conditions of Theorems 2 and 3, then, the independently adapting subset associated with an opposition mechanism contains all copies of all routes that are disjoint from the opposing set, while contiguous with a circuit into which the opposing set is embedded—that is, all routes fully opposed by the opposition mechanism in question. The Opposer Module presented in Fig. 1a,b then represents the class of network topologies that correspond to an independently adapting (or opposing) subset of an RPA equation.

Now, the hallmark of an RPA-capable network is the existence of a partition of its RPA equation into independently adapting subsets. (This general description includes the possibility of the trivial partition into a single subset comprising all of **R**). In addition, from the observation that the terms of **R** are distributed to independently adapting subsets by route (that is, all instances in **R** of a particular route are to be grouped together into a single such subset), it follows that these subsets are disjoint, and must cover **R**. We have seen, moreover, that two and only two mechanisms—which we call opposition and balancing—are able to generate the independently adapting subsets of **R** in an RPA-capable network, and that each such mechanism may be implemented by a rich class of sub-network topologies—Opposer Modules and Balancer Modules, respectively. Taken together, these considerations imply that a network can exhibit RPA only if it is decomposable into Opposer and/or Balancer Modules—that is, each route for the transmission of biochemical signal from input to output must be either balanced or (fully) opposed by a single network module.

A general RPA network could contain an arbitrary number of such modules—corresponding to its RPA equation being partitioned into (the same) arbitrary number of disjoint independently adapting subsets—so the question now remains as to how multiple such network modules may coexist (i.e., be connected together) in RPA networks.

**Interconnections of basis modules and larger RPA networks.** Now that we have a clear picture of the topological basis modules from which any RPA-capable network must be constructed, a set of rules governing how the modules may be combined

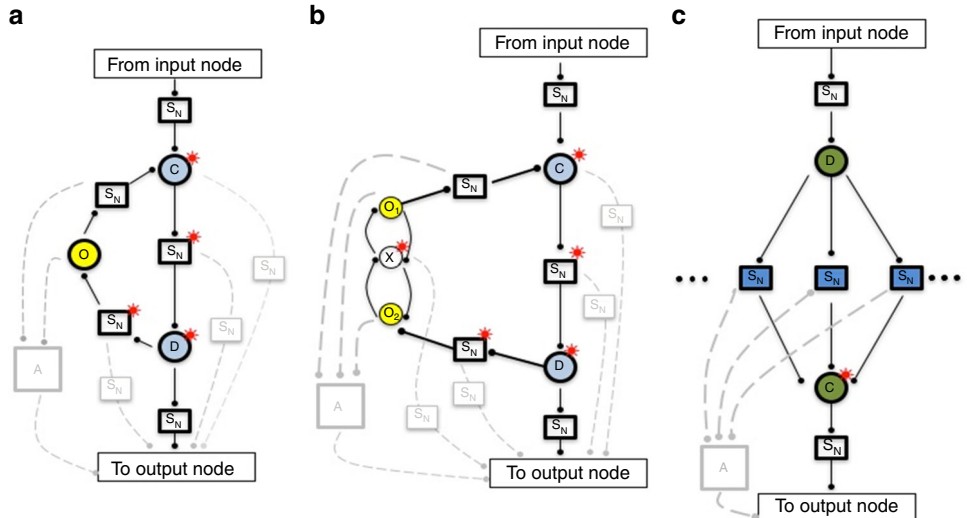

**Fig. 5** Outgoing regulations from RPA basis modules. The RPA basis modules from the point of view of potential outgoing regulations from each of the nodes that comprise the active part of the module. Any node that exhibits the RPA property (indicated with a red superposed asterisk) can produce a blind regulation: these can lead directly or indirectly to the output node without affecting the RPA-capability of the network. They can also regulate other RPA Basis Modules embedded in the larger networks without affecting the RPA-capacity of those modules. Any node within the module that does not exhibit the RPA property, on the other hand, can produce only a live regulation. Live outgoing regulations correspond to routes in the network that are not opposed/balanced by the module they feed out of. As such, they require an ancillary module to balance or oppose the live regulation in order for the network as a whole to be RPA-capable. **a**, **b** Opposer nodes cannot exhibit RPA. Any outgoing regulations from an opposer node (or any downstream dependents) place that opposer into a route that is not opposed by the associated Opposer Module. Such an outgoing regulation corresponds to term(s) of the RPA equation which belong to the complement of the independently adapting subset associated to the Opposer Module in question. As such, these outgoing regulations must feed into one or more ancillary RPA basis modules at the position(s) indicated by A. Blind regulations, on the other hand, correspond to terms of the independently adapting subset associated with the Opposer Module in question. These routes may thus be considered part of the module itself. **c** Balancer nodes do not exhibit the RPA property. Outgoing regulations from these nodes are necessarily live regulations, which correspond to the balancer node(s) in question participating in routes other than those that are balanced by the Balancer Module in question. These other routes are thus represented in the RPA equation in the complement of the independently adapting subset associated to the Balancer Module in question. As such, these routes must be balanced/opposed by one or more ancillary modules at the position(s) indicated by A

(interconnected) defines the full solution space of possible RPA-capable network topologies.

The nature of the two distinct RPA-generating mechanisms, and their topological realizations in self-organizing/self-regulating networks, does place some constraints on how RPA modules may be interconnected to form more complex multimodular networks. These constraints are two-fold: first, we note that the three types of reaction kinetics required to implement RPA—opposer kinetics, balancer kinetics and connector kinetics—are mutually exclusive (Supplementary Note 4). That is, any given node can exhibit at most one of the three types of reaction kinetics. Second, any given computational node (opposer, balancer or connector) is constrained in how it may be regulated: an opposer node, or a collection of collaborating balancer nodes, each has a single independent regulator; and a connector node works with a single collection of collaborating balancer nodes.

From this, we can conclude that the active part of each module (that is, nodes residing between the apex (node C in Fig. 1a, b and node D in Fig. 1c), and the base (D in Fig. 1a, b and C in Fig. 1c) must be distinct from the active part of any other module. A node that plays the role of an opposer in one module, for instance, cannot also be required to operate as a balancer (or a connector) for some other module. Moreover, the requirement for a single independent regulator implies that an opposer node can only perform its computational function for a single Opposer Module. Likewise, a set of collaborating balancer nodes, together with their connector node, delineates a single Balancer Module.

The requirement for distinctness of the active parts of RPA modules implies that the modules may either be connected 'in parallel', or 'in series' according to the definitions given in our Methods section.

**Live and blind regulations from intramodular nodes**. In order to make the series connection of RPA modules precise from a topological perspective, we first recall from the preceding sections that within the active part of each module, some nodes exhibit the RPA property, while others do not. Opposer nodes, along with any associated dependent nodes, do not exhibit the RPA property. The single independent regulator for an opposer, along with any associated dependent nodes, do exhibit the RPA property. Likewise, balancer nodes, along with their single independent regulator (the associated D-node) do not exhibit the RPA property, while connector nodes do exhibit the RPA property. From these considerations, we can consider any outgoing regulations from the active parts of an RPA module–leading ultimately to the network's output node—to be either 'blind regulations' if they come from node(s) that exhibit the RPA property or 'live regulations' if they come from node(s), which do not exhibit the RPA property.

We illustrate the essential principles of series interconnections of modules, which are required in any RPA network containing a module with live outgoing regulations, in Fig. 5. As shown in Fig. 5a, b, outgoing regulations from an opposer node (or associated dependent nodes) place that opposer in a route which must then be either balanced or (fully) opposed by some other RPA module (as indicated by the symbol A which indicates the position of the required ancillary module). Likewise, in a Balancer

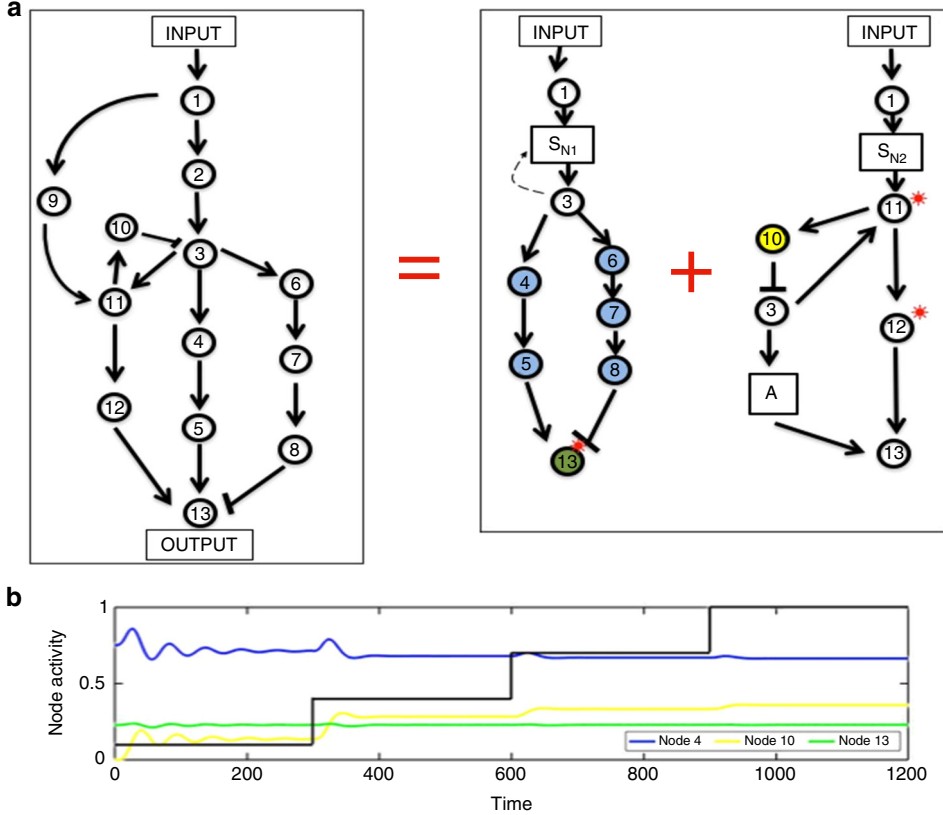

**Fig. 6** Opposition and balancing can collaborate in an RPA network. Any network that exhibits RPA must be decomposable into RPA Basis Modules. **a** Here we provide an illustration of a network topology that is capable of RPA via the decomposition into an Opposer Module (upstream) connected in series with a Balancer Module (downstream). The upstream Opposer Module is created if Node 10 (indicated in yellow) operates with opposer kinetics. Node 10 also participates in a route of the network, corresponding to a live outgoing regulation from the module at Node 3. This live regulation is balanced by the downstream Balancer Module when Nodes 4, 5, 6, 7, and 8 (indicated in blue) operate with balancer kinetics, and Node 13 (in green) operates with connector kinetics. Nodes 11, 12 exhibit the RPA property to the activity of the upstream Opposer Module; Node 13 (the network's output) then exhibits the RPA property due to the combined effects of the two Basis Modules. **b** Numerical simulations for a simple model of the network in **a**. Model equations and parameters are given in Supplementary Note 5. The input to the network is indicated in black

Module, outgoing regulations from balancer nodes place these nodes in routes that are not balanced by the module; these routes must be either balanced or (fully) opposed by some other ancillary module, as indicated by the symbol A in Fig. 5c. In either case, any outgoing blind regulations generate no requirements for any ancillary modules. Thus, any module with only blind outgoing regulations may exist alone in an RPA network, and any sub-network structures downstream of such regulations may be considered part of the module itself. In addition, blind outgoing regulations may feed into any other RPA module(s) without affecting the ability of those modules to contribute to RPA in the network as a whole.

Figures 6 and 7 present two different illustrative examples of RPA networks that are decomposable into two different RPA basis modules connected in series. In Fig. 6, the upstream module is an Opposer Module, whose single opposer node also contributes to two routes in the network (owing to the presence of live outgoing regulations from the opposer). These two routes are then balanced by the downstream Balancer Module. Figure 7 presents a network whose topological structure admits two different possible decompositions into RPA basis modules, depending on the choice of reaction kinetics at nodes 6 and 7. If the reaction kinetics at node 6 conform to opposer kinetics, for instance, this creates an Opposer Module where the single opposer node also participates in a collection of network routes; in this case, nodes 9 and 10 must be able to exhibit balancer

kinetics, and node 11 connector kinetics, in order for the network as a whole to exhibit RPA through the creation of a downstream Balancer Module (Solution 1). If node 7 were to operate with opposer kinetics, on the other hand, this would be sufficient to create a single Opposer Module from the entire network (Solution 2). Detailed analyses of these examples are given in Supplementary Note 5.

## Discussion

In the postgenomic era, as we continue to amass ever larger quantities of data on the vast and complex networks of molecular interactions within living systems, a tantalizing question continues to be raised: could complex biological systems be constructible from just a limited set of simple design principles?[20–26]. Here we show conclusively that, for RPA-capable networks at least, the answer is an unequivocal yes.

The centerpiece of the present work is the identification of two rich yet well-defined classes of network topologies which, together, span the space of all possible RPA-capable networks when suitably interconnected. These two classes of network module thus represent a topological basis for the solution to the RPA problem in any network, no matter how large or interconnected. In this sense, the topological basis modules are like the atoms of robust adaptation.

These findings represent a significant advance in our understanding of the basic structures underlying the complex and

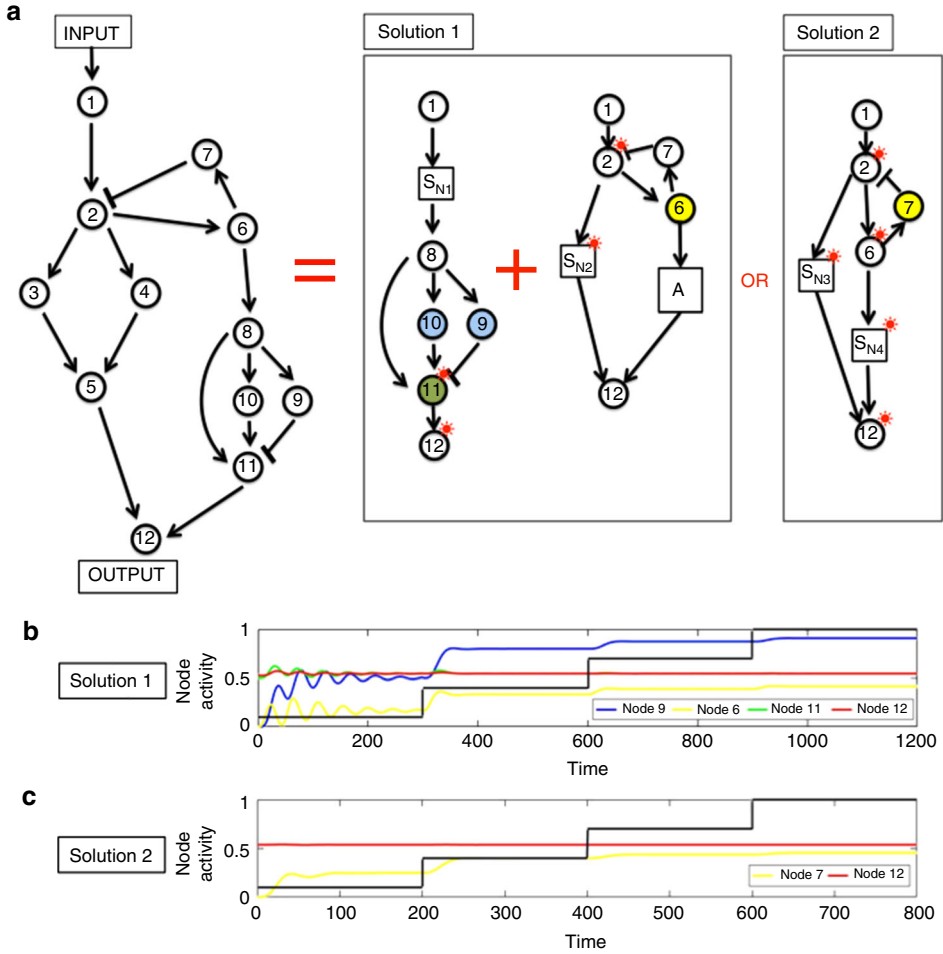

**Fig. 7** A single arrangement of nodes with multiple RPA solutions. Here we feature a network topology that is capable of exhibiting RPA through two different decompositions into RPA Basis Modules. **a** In Solution 1, the network is able to exhibit RPA as the series interconnection of an Opposer Module (upstream) with a Balancer Module (downstream). The upstream Opposer Module is created when Node 6 (indicated in yellow) exhibits opposer kinetics. Node 6 also participates in a route of the network, corresponding to a live outgoing regulation from the module at this node. This live regulation is then balanced by the downstream Balancer Module when Nodes 9 and 10 (indicated in blue) operate with balancer kinetics, and Node 11 (in green) operates with connector kinetics. Node 2 and its downstream dependents (Nodes 3, 4, and 5, indicated by $S_{N2}$ in the Solution 1 decomposition) all exhibit the RPA property due to the activity of the upstream Opposer Module. Node 11 exhibits the RPA property due to the activity of the downstream Balancer Module. Node 12 (the output node of the network) exhibits the RPA property due to the combined activities of the two RPA basis modules. In Solution 2, a single Opposer Module is created for the entire network when Node 7 (indicated in yellow) exhibits opposer kinetics. All nodes in the network (with the exception of Node 7 and Node 1) exhibit the RPA property due to the activity of this single RPA basis module. **b** Numerical simulations for a simple model of Solution 1. Model equations and parameters are given in Supplementary Note 5. The input to the network is indicated in black. **c** Numerical simulations for a simple model of Solution 2. Model equations and parameters are given in Supplementary Note 5. The input to the network is given in black

evolving networks occurring in nature. In many biological contexts—cellular signal transduction and cellular metabolism, for instance—the underlying signaling networks are so complex and high-dimensional, so prone to change over time, and so extraordinarily variable from one realization to another (even from one cell to another phenotypically identical neighboring cell), that the networks themselves are virtually impossible to define concretely at any useful level of detail. Although most investigators view this variability as a source of intractable complexity, particularly in our current age of Big Data, our work reveals that these networks may now be considered from the point of their unexpected simplicity—that is, as decompositions into well-defined basis modules.

It is interesting to consider these findings in the light of established results in control theory, which have determined that asymptotic tracking problems (such as RPA) require integral control as a structural property of the system[21–23]. Our work

shows that there are, in fact, two distinct types of integral control involved in the solution to the general RPA problem, corresponding to each of the two classes of RPA basis module: one type of integral is computed within feedback structures, employing specialized computational nodes (opposing sets) within collections of interlinked circuits; the other type is computed by a collaboration between two different types of computational nodes (balancers and connectors) embedded into parallel pathways (routes). Beyond this, we offer the novel insight that sufficiently large networks may solve the RPA problem via the arbitrary combination of the two topological basis modules, thereby distributing integrals of the two possible types throughout their vast assemblies of interacting nodes.

We summarize a wide selection of previously reported RPA examples[2,3,5–9,14] in Supplementary Note 7, and show that all such previous solutions are special cases of a single RPA basis module. All previous work to our knowledge considers networks

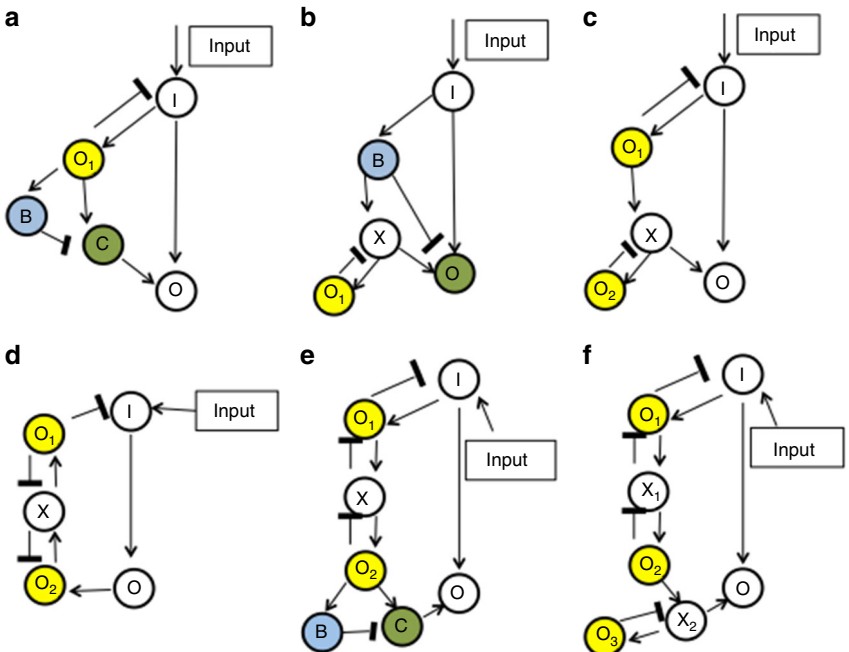

**Fig. 8** Small RPA networks with new topological features. We present several examples of the smallest network topologies that are capable of exhibiting RPA while incorporating several new topological possibilities uncovered in the present work. In particular, we examine RPA-capable network topologies that employ the combination of an Opposer Module and a Balancer Module, connected together (**a**, **b**, **c**). We also consider an example of the smallest network topology capable of implementing RPA using a non-trivial opposing set (**d**), and two examples of the smallest network topologies capable of implementing RPA via a non-trivial opposing set whose opposer node(s) also appear in a route, thereby requiring the collaboration of an ancillary module connected in series (**e**, **f**). Simple model equations for all six small network topologies, along with parameters and numerical simulations, are given in Supplementary Note 5, to confirm that these solutions do indeed engender RPA. **a** an Opposer Module (upstream) whose single opposer node ($O_1$, in yellow) also appears in a route of the network. This live outgoing regulation is balanced (downstream) by a Balancer Module. **b** a Balancer Module (upstream) whose balancer node B (in blue) also appears in a route that is not balanced by the module. This live outgoing regulation is opposed (downstream) by an Opposer Module. **c** A similar network construction to case **a**, except here the live outgoing regulation from opposer $O_1$ is opposed by a second Opposer Module connected in series downstream. **d** A single Opposer Module, employing the smallest non-trivial opposing set $\{O_1, O_2\}$ with associated master set $\{O_1, X, O_2\}$. **e** An Opposer Module with two-node opposing set (as in network **d**), whose opposer nodes also appear in routes of the network. This live outgoing regulation is balanced (downstream) by a Balancer Module. **f** A similar network construction to case **e**, except here the live outgoing regulation from opposer $O_2$ is opposed by a second Opposer Module connected in series downstream

that are so small and simple in construction that their RPA equations only admit the trivial partition into either a single S-set (generated by a single opposer node), or a single M-set (generated by a single balancer node and its collaborating connector). Indeed, all previously identified integral feedback in simple signaling motifs[2,3,5,8,16] are special cases of an Opposer Module using a single opposer node (itself a special case of an opposing set, Fig. 1a, b). The buffer nodes identified by Ma et al.[2] are special cases of opposer nodes. Previously identified examples of the incoherent feedforward motif[3,27–30] are special cases of the Balancer Module. The proportioner nodes described by Ma et al.[2] are special cases of balancer nodes.

We emphasize again that combinations of these two distinct mechanisms within a single RPA network have never before been proposed, presumably since the requisite network sizes are beyond the reach of blind computational screening methods.

In Fig. 8, we consider the smallest RPA networks that are capable of invoking the various novel topological features we identify in the present work, illustrating the significant increases in the sizes of the computational screening problems that would be required to identify these topologies. For a network to employ both an Opposer Module and an Balancer Module working together in collaboration, for instance, a minimum of five nodes would be required (Fig. 8a, b). Likewise, for a network to feature an opposer node that is also involved in a route (Fig. 8a, c), at least five nodes are needed. For a network with distinct input/

output nodes to incorporate a (non-trivial) opposing set (Fig. 8d), five nodes are, once again, the minimum requirement. If one or both of these opposer nodes are also in a route (Fig. 8e, f), the smallest such RPA networks contain seven nodes. Additional analysis of these small RPA topologies is presented in Supplementary Note 5.

Our general topological view of RPA networks highlights the role of antagonizing compensatory mechanisms—opposition and balancing—and the modular network structures induced by those mechanisms, in the robust regulation of signaling networks. This deep connection between compensation and modularity suggests that a modular design may characterize a wider class of robust networks beyond RPA-capable networks. Indeed, it raises the question as to whether all biochemical networks, of any size, with a fundamental need to exhibit robust functionalities, are characterized by modular architectures involving just a small number of topological basis modules.

Several lines of evidence already support the generalization of our network modularization to other bio-signaling contexts. The work of Averbukh et al.[31], and Ben-Zvi et al.[32], for example, point to a limited set of modules in the context of spatial signaling problems in embryonic development. Those authors undertook computational searches of small reaction-diffusion networks to identify configurations that could produce robust and scalable morphogen gradients[13,31,32]. Very few networks were capable of

generating the requisite robust patterning, and those that did were of a very specific type.

This paradigm shift suggests a resolution to a baffling paradox in living systems—that while networks of interacting molecules are often unimaginably complex, a property that is generally associated with fragility[1,20], such networks are nevertheless characterized by a remarkable robustness[1,17]. Indeed, robust traits are selected by evolution, and robustness facilitates evolvability in the face of changing and unfavorable environments. Our work provides strong evidence that a small number of universal modular designs offers a simplifying framework for the construction of complex bionetworks, giving them the capacity to scale to any size or complexity, without impairing their ability to adapt robustly, and without the requirement for any tuning of network parameters.

## Methods

### Methods overview and relationship to the IMP.
In the 1970s, Francis and Wonham[21,22] investigated the necessary controller structures required to achieve robust regulation with internal stability, and established what is now referred to as the IMP. By this principle, a controller can reject exogenous disturbances and/or track prescribed reference signals by incorporating within itself a model of the dynamic structure of the disturbances/references. More recently, Yi et al.[23] considered a special case of the IMP concerning RPA to constant exogenous inputs, in the context of providing a framework for understanding the extraordinary precision of adaptation in bacterial chemotaxis[5]. This analysis provided a purely algebraic condition that must be satisfied by an RPA-capable system, which has been shown to be equivalent to the requirement for integral control[23].

Here, our interest is not in confirming whether a particular network topology is capable of RPA, but in specifying all the possible network topologies—i.e., all the possible arrangements of nodes that are capable of exhibiting RPA, along with any constraints on the reaction kinetics for those nodes. For this, we begin by developing an alternative version of the algebraic condition specified by Yi et al.[23], for the special case in which a particular input/output node pair is specified, thereby constructing a framework from which topological structures may be deduced relative to that input/output node pair.

The generality of our method for studying RPA in biological networks, being self-organizing, self-regulating, complex, and evolvable, builds upon precise definitions of all the key terms of the problem, which we provide in detail in the attached Supplementary Information (SI). Briefly, we consider a node to be any entity that can encode and transmit a biochemical signal. Most commonly a node represents a molecule (its concentration, say, or the concentration of a particular activation state), a complex of molecules, or even a mathematical function of multiple biomolecular entities (See Supplementary Note 1). An input node is the recipient of some outside stimulus, $I$, while the end-point of interest (not necessarily distinct from the input node) is assigned the role of the output node. From this, a network may be defined, being the set of all nodes that are 'connected' and 'transmissive' relative to a chosen input/output node pair (see Supplementary Note 1 for detailed explanations of these terms). RPA is said to occur when the output node always returns to the same steady-state level, regardless of the magnitude of the stimulus delivered to the input node, with no requirement for special (fine-tuned) parameter choices.

Based solely on these general characteristics and definitions, for a network containing $n$ nodes $P_1,\ldots, P_n$, each with a reaction rate, $f_1,\ldots, f_n$, respectively, we show in Supplementary Note 2 that RPA occurs only when

$$\det(M_{IO}) = 0, \tag{1}$$

and

$$\det(J_n) \neq 0, \tag{2}$$

where $J_n = \frac{\partial(f_1,\ldots, f_n)}{\partial(P_1,\ldots, P_n)}$ is the $n \times n$ Jacobian for the system $\underline{f} = [f_1, \ldots, f_n]^T$, and $M_{IO}$ is the $(n-1) \times (n-1)$ input–output minor of $J_n$—that is, the matrix obtained by removing the input row and the output column from $J_n$, and where all matrix entries (i.e., partial derivatives) are evaluated at the network's steady-state, $\underline{\pi}_n = [P_1^*, \ldots, P_n^*]^T$ (see Supplementary Note 2 for detailed derivations and supporting discussion).

We refer to Eq. 1 as the RPA equation. In contrast to previous work[22], we do not consider this equation as simply an algebraic test for RPA; rather, we view the determinant expansion of the RPA equation from a topological perspective—that is, as a set of signed terms, together with collections of its subsets that may play independent roles in its solution. Noting that each term in the expansion contains topological information on the underlying network in terms of routes from input to output, feedback loops, and reaction kinetics, we thereby uncover general

principles as to how network sub-structures are able to work together in collaboration to generate RPA in arbitrarily large and complex networks.

It is clear that the RPA equation is a potentially huge equation in general, comprising some subset of the $(n-1)!$ terms corresponding to a fully-connected network of $n$ nodes. A 10-node network, for instance, could have as many as 362,880 terms in its RPA equation. Adding just five more nodes to give a 15-node network results in an equation of up to $8.7 \times 10^{10}$ terms. Doubling this network size to a 30-node network produces an RPA equation of up to $8.8 \times 10^{30}$ terms (see Supplementary Table 1). In any event, an arbitrary network of $n$ nodes will be able to exhibit RPA only if the $\tau \le (n-1)!$ terms of its RPA equation can sum to zero for all $I$ without violating Eq. 2.

We note that an alternative, but mathematically equivalent, version of the RPA equation has also been developed in the recent work of Tang and McMillen[33]. Those authors refer to the condition as 'the cofactor condition', and apply this approach to the issue of designing novel homeostatic systems. In particular, their design algorithm has been used to generate topologies and parameter constraints that 'will support homeostatic behavior for a given set of network components and a desired set of general regulatory constraints to be applied between them'[33].

### Deducing general mechanisms of RPA from the RPA equation.
We provide full details on our solution method for solving the RPA equation in complete generality in our Supplementary Information. As noted in the preceding section, Supplementary Notes 1 and 2 provide a complete set of precise definitions corresponding to our problem, and present detailed derivations of the RPA equation and the RPA constraint, along with mathematical forms for a set of axioms for the reaction kinetics at individual network nodes.

Supplementary Note 3 provides full details on our topological approach to the solution of the RPA problem, identifying conditions whereby the RPA equation may be partitioned into independently adapting subsets. To this end, we begin with the notion of a minimally adaptive subset of the RPA equation, of which there are two basic types—S-sets and M-sets—each with their own type of LAQ. We explore how these mathematical conditions imply novel topological structures within RPA networks—Opposer Modules (employing the novel concept of opposing sets) and Balancer Modules

The constraints on reaction kinetics that are imposed by the creation of S-sets and M-sets from the RPA equation are presented in detail in Supplementary Note 4. Here we also explore a range of parameter constraints that would allow the requisite reaction kinetics to be implemented in RPA networks, and also consider how these reactions implement some form of integral control.

Supplementary Note 5 considers the central matter of this work, namely the relationship between the two fundamental types of RPA Module (Opposer and Balancer) and a topological basis for RPA-capable networks. We consider how these topological basis modules may be interconnected to form larger (multimodular) RPA-capable networks.

To aid in the general delineation of all RPA-capable network topologies through the interconnections of Opposer and/or Balancer Modules we distinguish between the two possible relationships between interconnected modules, as we outline in the next section.

### Modules connected in series or in parallel.
Two RPA modules are said to be connected in parallel if none of the computational nodes within either module participate in route(s) that are opposed/balanced by the other. The respective route collections for the two modules must, therefore, diverge upstream of the active parts of the modules, and then reconnect again downstream of the active parts. Informally speaking, parallel modules are connected side-by-side within the global topology of the RPA network. When an opposer module is connected in parallel with all other RPA modules that comprise the network, for instance, its opposer node(s) do not participate in any route of the network; they participate in feedback loops only. This is a comparatively straightforward intermodular arrangement, then, for which we present an example in Supplementary Note 5 for two Opposer Modules connected in parallel (see Supplementary Figure 15).

A parallel arrangement of modules may be contrasted with the possibility that in some particular RPA module, one (or more) of its computational nodes may also participate in some network route that is not opposed or balanced by the module in question. For example, an opposer node—which operates within a feedback arrangement relative to the route(s) it opposes—may also participate in some route within the network. Likewise, a balancer node—embedded into the route segments defining its Balancer Module—may also participate in some other route in the network (that is, a route that is not balanced by the Balancer Module in question). In either case, the extramodular route(s) in which the computational node(s) participate must be either balanced or fully opposed by one (or more) additional RPA module(s) in series with the original module. Informally speaking, series modules are connected in an upstream-downstream arrangement, since computational nodes for the upstream module feed into the downstream module.

### Additional notes on Methods.
We conclude the detailed presentation of our methods in the Supplementary Information with a brief consideration of how all previously reported instances of RPA in the literature, to our knowledge, are special cases of the general solution we present in this article (Supplementary Notes 6 and 7).

For completeness, we also observe that although the topological structures we identify here are necessary conditions for solving the RPA problem in complete generality, these conditions are not sufficient by themselves to guarantee the implementation of RPA across all possible parameter regimes. In practice, RPA also requires global stability to ensure that there is a unique and stable steady-state regardless of initial conditions. We discuss stability issues briefly in Supplementary Note 8, where we point out that feedback loops, if present at all, should be negative-feedback loops since these are stability promoting. We nevertheless acknowledge that negative feedback could potentially induce oscillations or even chaotic behavior. More direct dynamical systems approaches are required to examine these possibilities for specific RPA topologies and specific parameter regimes.

**Code availability**. All computational simulations presented for illustrative purposes in this work were performed with MATLAB's inbuilt ODE solver, ODE45. All equations and parameters supplied to this solver are available in the Supplementary Information (see Supplementary Note 5).

**Data availability**. All data used in the present research are available upon request from the authors.

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

## Acknowledgements

This study was partially supported by NIH grants R33CA206937 and R01AR068436.

## Author contributions

R.P.A. conceived of the analytical methodology, and performed all derivations, proofs and computational simulations. R.P.A. and L.A.L. wrote the paper.

## Additional information

**Competing interests:** The authors declare no competing interests.

