## [Peer Review File · Nature Communications]

Reviewers' comments:

Reviewer #1 (Remarks to the Author):

I have mixed feeling about the manuscript "Only five classes of embedded motifs engender robust perfect adaptation in any network regardless of its size". On the positive side the authors made reasonable assumptions on the network regulation rules and based on which derived relationship between network topology and an important behavior called perfect adaptation. Relying on mathematical derivation instead of numerical method, the authors could explore a network that is much larger than people have been working on before. Based on Jacobian analysis the authors successfully derived several classes of networks that's capable of perfect adaptation. The authors described some interesting structure like multi-opposer control mechanism. From this perspective, I feel the author's efforts is very valuable. It expands our capability to deal with large regulatory networks.

But this manuscript is not without its problems. The general problems include: 1. Figures are not organized. Some of the main text figures don't show a conclusion by itself but works more like supplement figures to show examples instead. 2. There are many terms in different fields that the authors mixed together makes the article hard to read. 3. The main text is following a mathematical logic instead of a question driven logic line. This way the authors could lose many potential readers. 4. The authors also want to correct the color scheme of the time courses in figures, different colors are used on the output/input nodes which makes the figure hard to read.

The major problem of the manuscript is the classification of 5 classes of adaptation networks. The title is very attractive, but when I look through the main text and the supplement it is hard to find how the classification standard is chosen and why the way of classification is reasonable. Because this is the central point of the manuscript I would expect at least one figure to list all the five classes of motifs and explain the key features. There is also no such figure. Although figure 1 listed four of them but I have to go back and forth several time to realize figure 4 is the other one. When we really look into the 5 classes I am not sure it's the right way to categories all the networks. Generally, networks in figure one are all feedback loop controlled networks while figure 4 feedforward loops. For feedback controlled networks the author further divide them into 4 classes. I find this extremely confusion. First single-opposer classes could be special cases of multi-opposer classes. I think there should be at most two classes for S-set. These many classes just confuse people. Second all the opposer sets should use integral control as the mechanism which makes the feedforward loop dispensable. This means only one class is absolutely necessary to represent all the networks in the S-set. But actually if there are feedforward loops and feedback loops co-existing in one network the adaptation could due to either feedback control (S-set control) or feedforward control (M-set control), based on different parameters. The authors seem to totally neglect the importance of control parameters. All in all, I am not convinced the conclusion in this manuscript is sound and helpful to the field although I think the way the authors approach the problem is valuable.

Reviewer #2 (Remarks to the Author):

I'm sympathetic to the aims of this paper but it needs major revisions to achieve them.

Details of review for NCOMMS-17-05100

This paper aims to expand and refine the concept of Robust Perfect Adaptation (RPA) as it arises in cell biology. The aims are interesting and even important, but I think the paper as written falls short. However, if the details were corrected and improved, the author's aims would be strengthened. So this review is critical of the technical details, but I'm sympathetic to the aims and would like to see a paper like this succeed. I'll sketch what the main aims are and then go into selected details that I think are the best opportunities for improvement. As I'm sympathetic to the aims of the paper, and the authors have clearly done a lot of work, I spent a lot (too much in retrospect) time going through the details in the supplement for anything that would justify the proliferation of definitions and mitigate the main problems focused on below. Unfortunately, I was unsuccessful, but the supplement is so long and involved it is hard to be certain. The highlights:

Necessary conditions for RPA:

"... in stark contrast to previous work on the RPA problem (see (3,18) for recent reviews), we present a global and 'top-down' methodology that allows us to identify the necessary conditions for achieving RPA in arbitrarily large and complex networks." (see line 57) The "previous work" cited is all recent papers in biology, where the literature is admittedly a bit of a mess. What is overlooked is a much bigger and older (>40 years) literature in engineering and math in control theory. One centerpiece of this paper, the "RPA equation" is actually a special case of a special case of the Internal Model Principle (by Francis and Wonham, 1977). A very brief primer with this and additional reference is in

Yi, Huang, Simon, Doyle: (2000) Robust perfect adaptation in bacterial chemotaxis through integral feedback control. *Proc. Natl. Acad. Sci.* 2000: Apr 25 97(9): 4649-53

Yi et al appears to be the only biology paper like this, but it uses only the most trivial parts of control theory that was already decades old in 2000. The RPA equation is trivially equivalent to the RPA conditions given in Yi et al. I'll expand on this in more detail below.

Five and only five classes

Beyond the RPA equation, a central claim in this paper is that (line 86) "...five and only five distinct classes of network modules represent a basis for the solution to the RPA problem in any size network." The aim here is interesting but the claim appears not to be correct. The 5 classes are in fact only 2 in disguise, though the disguises are quite clever. I'll work through the details of this below, and the required machinery is vastly simpler than what is in the paper.

Minimal modules

The authors also seem to have overlooked a crucial minimal solution in their list of "minimal modules" in S6.2 (page 47). Below I'll describe what I find to be the two "fundamental" motifs. This has all been well-known in engineering for decades, but is no longer emphasized and would deserve some revival in the context of biology. The paper would be much clearer if such minimal modules were corrected and emphasized more rather than devote many pages to algebra for what seem to be unnecessarily complex case studies.

Primer on integral feedback control theory basics: The paper mentions integral feedback control (IFC) but never clearly states what it is, though it is implicit in much of the supplement. Since IFC is such a simple concept it is probably worth explicitly mentioning what it is. The RPA equation, a centerpiece of this paper, is a simple special case of the IFC *det* condition below, which is a special case of the Internal Model Principle (IMP). The paper incorrectly claims that the RPA equations is a “generalization of integral control” (191) when it is a special case of a special case of a result (IMP) that is now >40 years old. For references on IMP and a brief primer on Integral Feedback Control (IFC) for biologists, see Yi et al above.

I’ll quickly review below what is in Yi et al 2000 *PNAS* paper but note it has been “well known” (taught to undergrads) regarding “perfect adaptation” in engineering for at least 30 years. The current submission’s main results are largely special cases, so the paper would be both more scholarly and clearer if it adopted these standard results. The confusion arises on line 57 where the paper states that “...until now, three basic approaches have been used to understand the requirements for RPA (2,3,5,8,14,18,19) , and they have only provided answers for very small systems.” The authors claim that “... in stark contrast to previous work on the RPA problem (see (3,18) for recent reviews), we present a global and ‘top-down’ methodology that allows us to identify the necessary conditions for achieving RPA in arbitrarily large and complex networks.” If you ignore the Yi et al *PNAS* paper above, this is largely true in the biology literature but has emphatically not been true for >40 years in undergrad engineering control theory. What follows is a minimal primer on this material.

Starting simply, consider the linear system

$$\begin{bmatrix} \dot{x} \\ y \end{bmatrix} = \begin{bmatrix} A & b \\ c & d \end{bmatrix} \begin{bmatrix} x \\ w \end{bmatrix} = \begin{bmatrix} A & b \\ c & 0 \end{bmatrix} \begin{bmatrix} x \\ w \end{bmatrix}$$

where y and w are scalar output and input, and x is a vector internal state and all vectors and matrices are real and of compatible dimensions. It is actually simpler to use this general case rather than the special case where some states are identified as inputs and/or outputs. We’ll assume that $c \neq 0$ for nontriviality (and we are assuming $d=0$ as in the paper, not an important restriction).

We’ll consider some arbitrary initial condition in x and a step input in w , both at $t=0$. Then “perfect adaptation” (PA) is defined to be the property that

$$\text{PA: } y(t) \xrightarrow[t \rightarrow \infty]{} 0 \quad \forall x(0) \text{ initial conditions} \quad \forall w(t) = \text{constant (step)}$$

This goes by other names in engineering (asymptotic tracking, zero steady state, ...) but we’ll use the biologist’s PA. For *Robust* PA (RPA) we need to define some uncertain set $[A, b, c] \in S$ and check that PA holds for all $[A, b, c] \in S$. We’ll assume A is stable throughout. Stability is largely ignored in this paper, and discussed briefly in S9 (Supplement Section 9). More on this later, but note that for any nontrivial uncertainty model, verifying robustness of PA is trivial compared with robustness of stability, and thus not surprisingly robust control theory focuses on the latter. In biology the relative simplicity of testing RPA should probably be a feature.

The main “integral feedback control” (IFC) theorem is

IFC Thm:

$$PA \Leftrightarrow \det \begin{bmatrix} A & b \\ c & d \end{bmatrix} = 0 \Leftrightarrow \exists k \ni z = kx \quad \& \quad \dot{z} = y$$

This is easily proven with a few lines of algebra, given in the Yi et al paper. This is arguably the simplest nontrivial result in control theory and is a special case of the Internal Model Principle (IMP), that PA to an external disturbance like a step requires an internal model. For example, PA to steps, ramps, and sinusoids need, respectively, an internal single and double integrator, and an oscillator. Let me underscore here that the “RPA equation” is strictly a special case of the det condition in the IFC Thm (easily shown by simply taking the special case in the paper and applying the Schur complement formula to the determinant). In the author’s defense these simple results are perhaps not adequately emphasized in control theory education, and typically mentioned mostly in passing, as RPA seems to be such a trivial special case that it doesn’t need highlighting. I think this attitude makes sense in engineering but perhaps less so in biology. In any case, biologists are clearly interested in RPA so it would help them to have a rigorous and accessible account.

The integral feedback is in the variable $z = kx$ which has dynamics $\dot{z} = y$. This is popular in engineering because it is easily implemented in digital controllers, and the main technical issue is called “integrator windup” which occurs with saturating actuators, and thus integral controllers include antiwindup mechanisms. In biology, the issue is more “reverse engineer” to identify mechanisms that achieve PA, and the paper cites several of the standard papers in this genre. The *det* condition is a simple algebraic test for PA and is equivalent to the existence of integral control.

Beyond the RPA equation, a central claim in this paper is that (86) “... fundamental solutions to the RPA problem is limited to five: that is, five and only five distinct classes of network modules represent a basis for the solution to the RPA problem in any size network.” I was unable to find a clear definition of what “fundamental solutions” are but it seems fairly clear to me that their five distinct classes are in fact elaborations of just two “fundamental” motifs.

The authors also seem to have overlooked a crucial minimal solution in their list of “minimal modules” in S6.2 (page 47). I’ll next describe what I find to be the two “fundamental” motifs, the first of them equivalent to S6.2.1 Case A on page 49, but the second one is a nonlinear “balancer” module that is simpler than S6.2.3 Case C on page 49. This has all been well-known in engineering for decades, but is no longer emphasized and would deserve some revival in the context of biology.

Minimal “motifs”: Both minimal motifs have 2 states. The first is the pure example of integral feedback control (IFC) and can be used to illustrate a simple but meaningful notion of robustness. Consider the system

$$\text{Case 1 (Minimal IFC): } \left[\begin{array}{c|c} A & b \\ \hline c & d \end{array} \right] = \left[\begin{array}{cc|c} -1 & -1 & 1 \\ 1 & 0 & 0 \\ \hline 1 & 0 & 0 \end{array} \right] \Rightarrow \det \begin{pmatrix} A & b \\ c & 0 \end{pmatrix} = 0 \Rightarrow z = x_2 \quad \& \quad \dot{z} = y$$

This also has a variety of robustness features. For example consider the same sparsity pattern

$$\left[\begin{array}{c|c} A & b \\ \hline c & d \end{array} \right] = \left[\begin{array}{cc|c} -a_{11} & -a_{12} & 1 \\ a_{21} & 0 & 0 \\ \hline 1 & 0 & 0 \end{array} \right] \quad \forall a_{ij} > 0 \Rightarrow \det \begin{pmatrix} A & b \\ c & 0 \end{pmatrix} = 0 \Rightarrow z = x_2 \quad \& \quad \dot{z} = a_{21}y$$

Note that the det condition (and stability) holds for all $a_{ij} > 0$ and the integrator is trivially in the 2nd state. This is the purest example of RPA and all but one of the “5 distinct classes” are simply

elaborations of this where the pure integrator is replaced by a circuit that implements an integrator. More details on this later.

A second case which biologists call Incoherent Feedforward (IFF) has a linearization of

$$\text{Case 2 (Minimal IFF): } \left[\begin{array}{c|c} A & b \\ \hline c & d \end{array} \right] = \left[\begin{array}{cc|c} -1 & 0 & 1 \\ -1 & -1 & 1 \\ \hline 0 & 1 & 0 \end{array} \right] \Rightarrow \det \begin{pmatrix} A & b \\ c & 0 \end{pmatrix} = 0 \Rightarrow z = [1 \quad -1]x \quad \& \quad \dot{z} = y$$

but here the integrator is a mixture of the 2 states. This is not as robust as the first case, in that

$$\left[\begin{array}{c|c} A & b \\ \hline c & d \end{array} \right] = \left[\begin{array}{cc|c} -1 & 0 & 1 \\ -a_{21} & -1 & b_2 \\ \hline 0 & 1 & 0 \end{array} \right] \Rightarrow \det \begin{pmatrix} A & b \\ c & 0 \end{pmatrix} = a_{21} - b_2$$

So that it appears that the parameters must be “fine-tuned” so that $a_{21} = b_2$. However, a nonlinear version can be easily made robust, and the authors seem to have rediscovered this, though made it unnecessarily complicated. Consider the nonlinear system with input w and output P (using notation similar to the paper) and additional state B

$$\begin{aligned} \dot{B} &= (1+w) - B \\ \dot{P} &= \alpha(1+w) - BP \end{aligned} \quad \Rightarrow \quad \left\{ \begin{array}{l} \text{Steady state} \\ \dot{B} = \dot{P} = 0 \end{array} \right\} \Rightarrow \begin{aligned} B &= (1+w) \\ P &= \alpha \end{aligned}$$

so the output P has RPA with respect to the parameter α . The linearization around $w=0$ is a version of Case 2:

$$\left[\begin{array}{c|c} A & b \\ \hline c & d \end{array} \right] = \left[\begin{array}{cc|c} -1 & 0 & 1 \\ -\alpha & -1 & \alpha \\ \hline 0 & 1 & 0 \end{array} \right] \Rightarrow \det \begin{pmatrix} A & b \\ c & 0 \end{pmatrix} = 0$$

So the linearization appears to need fine-tuning but since $a_{21} = b_2 = \alpha$ this is done “for free” by the linearization of the nonlinear system. This has been viewed primarily as a curiosity in engineering, but it appears from earlier work in biology that this nonlinear motif does arise natural in biology and deserves further study. In the paper this case study is not identified as one of the “minimal” modules but is part of S6.2.3 Case A which has 3 additional (and unnecessary) states.

While the authors seem to overlook the minimality of Case 2, they would agree with this analysis that Case 1 and 2 are distinct cases. Where we would appear to disagree is that I can easily show that all of their 5 “distinct cases” are in fact nearly trivial elaborations of these two “motifs”. By “nearly trivial elaborations” I mean that their cases can all be shown equivalent to starting with Case 1 or 2 and then adding dynamics that can affect stability but not RPA, so these additional dynamics are inconsequential for RPA. I’ll work through one case study at the end.

So it is easily shown that the 6 minimal modules in S6.2 (pages 47-50) are not minimal. In particular, S6.2.3 Case C is an elaboration of Case 2, and the rest are either exactly Case 1 (S6.2.1 Case A) or are elaborations of it (the rest). This is easily shown constructively by simply using the IFC Theorem to extract the z variable that is the integrator, and the rest of the states don’t contribute to PA and merely need to preserve stability. To concretely illustrate this, next I’ll show how S6.2.4 Case D on page 49 is a trivial elaboration of Case 1 above and is not a “distinct class of network module” from Case 1. The same argument applies to 4 of the 5 classes, with the remaining class being equivalent to Class 2 above.

The nonlinear version of Case 2 has an integrator in its linearization but as a nonlinear system is not equivalent to Case 1, so Cases 1 and 2 are legitimately 2 distinct “motifs” that can yield RPA, but the remaining 3 “classes” are not distinct in any meaningful sense. But to make this clear we will carefully work through S6.2.4 Case D and show how it almost trivially reduces to Case 1 above.

Showing S6.2.4 Case D is a version of Case 1:

Recap Case 1 and add a simple notation for the motif, which we’ll call the Minimal IFC motif:

Case 1: $\left[\begin{array}{c c} A & b \\ \hline c & d \end{array} \right] = \left[\begin{array}{cc c} -1 & -1 & 1 \\ 1 & 0 & 0 \\ \hline 1 & 0 & 0 \end{array} \right]$	Motif:  Minimal IFC
--	---

Now consider S6.2.4 Case D which has RPA and can be mostly simply written and drawn this way

S6.2.4 Case D: $\left(\begin{array}{c c} A & b \\ \hline c & d \end{array} \right) = \left(\begin{array}{cccc c} -1 & 0 & 0 & 1 & 1 \\ 1 & 0 & -1 & 0 & 0 \\ 0 & 1 & -1 & 1 & 0 \\ \hline 0 & 0 & -1 & 0 & 0 \\ 1 & 0 & 0 & 0 & 0 \end{array} \right)$	Motif: 
--	---

We can check the conditions for the IFC Theorem:

$$A \text{ stable} \quad \det \begin{bmatrix} A & b \\ c & d \end{bmatrix} = 0 \Rightarrow$$

$$\text{Integral feedback: } z = [0 \quad 1 \quad 0 \quad -1]x \quad \dot{z} = y$$

And then change coordinates so that z is the new 2nd state, and then redraw the motifs.

Change coordinates: $\left(\begin{array}{c c} T & 0 \\ \hline 0 & 1 \end{array} \right) = \left(\begin{array}{cccc c} 1 & 0 & 0 & 0 & 0 \\ 0 & 1 & 0 & -1 & 0 \\ 0 & 0 & 1 & 0 & 0 \\ \hline 0 & 1 & 0 & -1 & 0 \\ 0 & 0 & 0 & 0 & 1 \end{array} \right) \rightarrow$	$\left(\begin{array}{c c} TAT^{-1} & Tb \\ \hline cT^{-1} & d \end{array} \right) = \left(\begin{array}{cccc c} -1 & -5 & 0 & .5 & 1 \\ 1 & 0 & 0 & 0 & 0 \\ 0 & 0 & -1 & 1 & 0 \\ \hline 1 & 0 & -2 & 0 & 0 \\ 1 & 0 & 0 & 0 & 0 \end{array} \right) \downarrow$
---	---

What we see here is that after a simple change of coordinates, S6.2.4 Case D is a special case of a more general motif that is a simple elaboration of the Minimal IFC. The truly general case is this, where the $\det=0$ condition is trivially seen by inspection of the matrices:

My point is that this is a huge class of networks all of which have RPA and are an obvious elaboration of the Minimal IFC. By “elaboration” I mean adding dynamics that can affect stability but not RPA (so must preserve stability), but these additional dynamics are inconsequential for RPA. The other cases in Section 6 are more complicated but similar. For example, this motif is an obvious elaboration of the Minimal IFC (see also 6.2.2 Case B), but has a different output.

Note that 4 of the 5 “distinct classes” are just such elaborations, with added dynamics that have no effect on RPA, which only depends on the existence of a single integrator state (which may be a linear combination of states in the original coordinates). This does not rule out that there may be more “distinct classes” to be found (the elaboration of the IFF case 2 is less obvious), but the one in the paper do not qualify.

Note also that the direct use of the IFC theorem vastly simplifies the arguments here, certainly compared with the complexity of those in the supplement. Even if corrected, the results simply don’t seem to need this level of complexity. RPA and the IFC theorem is arguably the simplest nontrivial result in control theory, and deserves more attention, particularly in biology, but not greater complexity unless that complexity buys additional insights.

Response to Reviewers (R.P. Araujo and L.A. Liotta; NCOMMS-17-05100).

Response to Reviewer #1

We warmly thank this reviewer for such supportive input on our work. We have copied the entirety of the reviewer's report below (*in italicized type, and indented*), and have followed each comment or criticism with our response, including an explanation as to how we have changed our manuscript.

By way of a general overview of our extensively revised manuscript, we note this reviewer's major technical concern that the original categorization of the modules into **five** distinct types may have introduced unnecessary distinctions among the modules. After careful consideration and analysis of this matter, we completely agree with the reviewer's assessment and find that we can indeed express the full solution space of RPA network topologies in terms of **just two** classes of basis modules under our new unifying definitions. One class is the "opposer" modules (orchestrated by the "opposition" mechanism, and for which we had previously, and needlessly, subdivided into four different categories), The second class is the "balancer" modules (orchestrated by the "balancing" mechanism). In particular, we have concluded that the reviewer is entirely correct in suggesting that single opposers may be considered special cases of opposing sets. In addition, we have now discovered a precise way to describe "*feedforward*" *opposition* (where opposer nodes, acting in feedback formation, *also occur in a route* and thus contribute a transmissive, feedforward, function in the network) in terms of the *interconnectivity* of modules, rather than incorporating this into the module itself as a topological feature.

The centrepiece of this work is the identification of network elements that can truly be considered **basis elements**, along with a **general way to combine** (interconnect) those elements, so as to **span the complete solution space** to the RPA problem. Remarkably, grouping our basis modules into just two classes (rather than five), and incorporating the novel phenomenon of feedforward opposition into the interconnectivity of modules has tremendously simplified our communication of the general RPA solution. Our descriptions of both the modules themselves, and their interconnections, have now become vastly more streamlined and elegant, and present to the reader a more accessible view of the overarching structures governing robustness in ALL complex networks.

The reviewer has also made several other valuable suggestions for the improvement of our presentation, for which we are truly grateful. We have taken careful note of all these helpful comments, as we note in the following point-by-point responses:

Reviewer #1 (Remarks to the Author):

I have mixed feeling about the manuscript "Only five classes of embedded motifs engender robust perfect adaptation in any network regardless of its size". On the positive side the authors made reasonable assumptions on the network regulation rules and based on which derived relationship between network topology and an important behavior called perfect adaptation. Relying on mathematical derivation instead of numerical method, the authors could explore a network that is much larger than people have been working on before. Based

on Jacobian analysis the authors successfully derived several classes of networks that's capable of perfect adaptation. The authors described some interesting structure like multi-opposer control mechanism. from this perspective, I feel the author's efforts is very valuable. It expands our capability to deal with large regulatory networks.

We thank the reviewer for this generous support and encouragement, and for the recognition of various novel and unexpected aspects of both our method and our findings. As the reviewer points out, our mathematical view of the RPA problem (in comparison with previous, predominantly computational, approaches) has been able to identify topological features of robust networks that have never been observed before – features such as “opposing sets” involving multiple opposers working together in collaboration as a complex control structure. The reviewer also acknowledges that our work establishes a connection between *network function* (RPA, in this case) and *network topology*, and that our approach allows us to explore much larger regulatory networks than have ever been considered previously.

But this manuscript is not without its problems. The general problems include: 1. Figures are not organized. Some of the main text figures don't show a conclusion by itself but works more like supplement figures to show examples instead. 2. There are many terms in different fields that the authors mixed together makes the article hard to read. 3. The main text is following a mathematical logic instead of a question driven logic line. This way the authors could lose many potential readers. 4. The authors also want to correct the color scheme of the time courses in figures, different colors are used on the output/input nodes which makes the figure hard to read.

We thank the reviewer for this constructive feedback and for the opportunity to significantly improve the presentation of our work. Our response, point-by-point, is as follows:

1. The reviewer's comment that our figures could be better organised is well taken. In particular, the reviewer suggests that “*Some of the main text figures don't show a conclusion by itself but works more like supplement figures to show examples instead.*” We have especially taken careful note of the reviewer's view (expressed later in his/her report) that at least one figure should show all the classes of modules and explain the key features. With these helpful comments in mind, we have completely rethought and revised the contents of our figures so that they are more illustrative of the central conclusions of our work, and better support the communication of our main arguments. Our extensively revised manuscript now comprises a total of eight brand new figures, whose main features we summarize below:
 - a. Figure 1 summarises the two distinct classes of RPA basis modules. For the sake of clear illustration, two different examples of opposer modules are shown – one with a single opposer node (which we now consider to be the “trivial” opposing set) in part (a), and one with the smallest non-trivial opposing set (ie. a two-node opposing set) in part (b). Part (c) of the figure gives a general representation of the balancer module. Here (and in all the subsequent figures), use small superposed red asterixes to indicate all nodes within the module that exhibit the RPA property.
 - b. Our new Figure 2 now clarifies the role of “integral control” in the solution to the RPA problem. The two different “mechanisms” of RPA – namely, opposition and balancing – are implemented via distinct sets of computational nodes (opposer nodes, for opposer modules; and the

collaboration between at least one balancer and a connector node, for balancer modules). This figure makes clear how these two mechanisms play distinct computational roles in the solution of the RPA problem, through the computation of distinct types of integrals. For the sake of clear illustration of the basic principles, we have chosen the most “minimal” version of each type of module, and the simplest type of appropriate reaction kinetics (for the opposer, balancer and connector). Again, we indicate the nodes that exhibit the RPA property within the module with a superposed red asterisk.

- c. Since our work is the first to identify and name “Opposing Sets” as a topological possibility in larger RPA-capable networks, our new Figure 3 gives several different examples of how these well-defined arrangements of opposer nodes could be embedded into a feedback structure. As we clarify in the caption to this figure, our Theorem 3 (SI) specifies all possible such arrangements of opposer nodes into collections of interlinked circuits. The figure uses three different realizations of this principle to communicate to the reader how the members of the opposing set work together to compute the integral of some “tracking error” for a particular node in the route(s) that they collectively oppose. These illustrations highlight the crucial aspect and that opposing set’s ability to accomplish this feat hinges upon the ability of each individual opposer node in the set to compute the integral of a tracking error in one particular node within the overall topological arrangement. The consequence of this “team effort”, as indicated, is that various other nodes within the modular topology (in addition to the route node) will also exhibit the RPA property (indicated, according to our chosen convention, with a superposed red asterisk). The distinctive topological feature of opposing sets, is that they participate in feedback loops that are disjoint from the route(s) they collectively oppose.
- d. Figure 4 depicts a more general representation of the essential structure of opposing sets, along with their topological relationships to the route(s) they oppose. As the reviewer has noted, the discovery of opposing sets as sub-network topologies that can be embedded within RPA-capable networks is an important novel finding of our work.
- e. A major improvement to our revised submission, thanks in no small part to the reviewer’s (correct) suggestion that the completely general RPA solution should be expressible in terms of just two classes of modules, is how we have been able to communicate the interconnections between modules in larger (multimodular) networks. Figure 5 emphasizes the role of the possible outgoing regulations from the two classes of modules, and indicates that these fall within two major categories: “blind” regulations, which are outgoing regulations RPA-exhibiting nodes within the module; and “live” regulations, which are outgoing regulations from nodes that do not exhibit the RPA property. In our revised submission we now show that “live” outgoing regulations correspond to routes in the network that must be either opposed or balanced by at least one additional ancillary module. Such an ancillary module must be connected into the network at the position indicated “A” indicated in Figure 5. As we explain within the text of the manuscript, the ancillary module(s) are connected “in series” with the

original module. The new concept of “live “ and “blind” regulations, we believe is another exciting outcome of the simplification of the module number and its translation to large, previously unapproachable, networks.

- f. One of the key achievements of our work is that we are able to explain, in a general way, how individual RPA modules may be connected together to form larger RPA-capable networks. In Figure 6, we illustrate the general principles through an example in which an opposer module (using a single opposer node) is connected “in series” with a balancer module. Here again, we indicate which nodes exhibit the RPA property with a superposed red asterisk.
- g. One of the reviewer’s extremely valuable suggestions (indicated later in his/her report) was that we should explain more clearly the role of different parameter regimes (which are able to produce the different classes of reaction kinetics), and to highlight the possibility that a given overall network structure could admit different RPA solutions, depending on the types of reaction kinetics that are implemented at the various nodes. (*“ ... if there are feedforward loops and feedback loops co-existing in one network the adaptation could due to either feedback control (S-set control) or feedforward control (M-set control), based on different parameters. The authors seem to totally neglect the importance of control parameters.”*) In Figure 7, we have presented an example of a network that does indeed admit two different RPA solutions, depending on the reaction kinetics at certain key nodes. Thus, although the major conclusion of our work is that all RPA networks must be decomposable into RPA basis modules, we show here that a given network construction could, in principle, admit multiple distinct decompositions into basis modules. We sincerely thank the reviewer for this helpful and insightful suggestion, as it does give us the opportunity to clarify an important technicality, and gives the reader greater insight into the possible structures of RPA-capable networks.
- h. In Figure 8, explore the issue of how many nodes are required to incorporate various topological features that our work reveals for the first time. As we point out in our revised manuscript:
“In Figure 8, we consider the smallest RPA networks that are capable of invoking the various novel topological features we identify in the present work, illustrating the significant increases in the computational screening problems that would be required to identify these topologies. For a network to employ both an opposer module and a balancer module working together in collaboration, for instance, a minimum of five nodes would be required (Figure 8a,b). Likewise, for a network to feature an opposer node that is also involved in a route (Figure 8a,c), thereby requiring an additional ancillary module connected in series, at least five nodes are needed. For a network with distinct input/output nodes to incorporate a (non-trivial) opposing set (Figure 8d), five nodes are, once again, the minimum requirement. If one or both of these opposer nodes are also in a route,

however, thereby requiring the collaboration of an ancillary module (Figure 8e,f), the smallest such RPA networks contain seven nodes.”

2. The reviewer suggests that *“There are many terms in different fields that the authors mixed together makes the article hard to read”*. Although the reviewer is not terribly specific on this point, **we have completely rewritten our manuscript, and have carefully revised our Supplementary Information document**, and have taken great pains to streamline our explanations, and present a more clear and lucid account of our key findings, in order to make our work as accessible and interesting as possible to as large a potential readership as possible. We trust the reviewer feels that this aspect of our work is much improved, and thank the reviewer for his/her patience and care in considering our work so carefully.
3. The reviewer also suggests that *“The main text is following a mathematical logic instead of a question driven logic line. This way the authors could lose many potential readers.”*. Once again, we emphasize that we have completely rewritten our manuscript to make our work clear and engaging, and to emphasize to the reader that our novel findings represent a paradigm shift in our understanding of the basic structures underlying the complex networks occurring in nature. We also emphasize how our work provides fundamental questions to some of the most perplexing questions in molecular biology; for instance, **“During development, and throughout evolution (1), biologic networks - or “bionetworks” - grow to enormous size and biochemical complexity, apparently without any compromise in robustness. Why isn’t this growth in complexity associated with heightened fragility, or instability, or a loss of requisite function (18-20)?**

We go on to explain that,

“ ...in this connection, it is essential to recognise that biological systems differ in fundamental ways from engineering control systems. Molecular signaling networks are self-organising, self-regulating and evolvable, and as such, are comprised of elements that must serve both as the transmitted signals and their own controllers. Unlike their designed counterparts in engineering control systems, bionetworks do not have the luxury of employing specially-designed, dedicated components whose purpose is to sense or control biochemical signals. Although asymptotic tracking problems (of which RPA to constant exogenous inputs is a special case) have been studied extensively for engineering systems (21), how can we understand the mechanisms governing robust performance in the context of the self-organizing, self-regulating autonomous systems arising in biology?

“In particular, how can RPA-generating mechanisms be realized topologically in bionetworks – that is, in terms of the arrangements of interconnected nodes, along with the types of chemical reactions regulated by the interacting nodes? Do the

topological requirements for RPA in large systems increase in complexity, or change qualitatively, along with the growth of the system, or do larger systems simply replicate the same basic design principles used by smaller systems? If the latter is the case, what are the universal topological principles that characterize general RPA-capable network designs (18)?

We go on to point out in our Discussion section that,

“In many biological contexts – cellular signal transduction and cellular metabolism, for instance – the underlying signaling networks are so complex and high-dimensional, so prone to change over time, and so extraordinarily variable from one realization to another (even from one cell to another phenotypically-identical neighboring cell), that the networks themselves are virtually impossible to define concretely at any useful level of detail. Whereas most investigators view this variability as a source of intractable complexity, particularly in our current age of Big Data, our work reveals that these networks may now be considered from the point of their unexpected simplicity – that is, as decompositions into well-defined basis modules. In this sense, complex bionetworks are like snowflakes: while each is unique, all individual instances are alike in their essential structure.

Throughout our revised manuscript, we make every effort to clearly identify the relationship of our methods and our findings with these fundamental questions and problems.

4. The reviewer suggests some improvements to the color schemes in the time courses in some of our figures, explaining that “different colors are used on the output/input nodes which makes the figure hard to read.” As we mention in our response to Point 1 above, we have now completely revised the figures for our revised manuscript, and have taken the utmost care to ensure that the color and notation conventions are consistent throughout.

The reviewer then moves to his/her central criticism, concerning whether five classes are really needed in order to adequately describe RPA networks in sufficient generality, or if just two classes (for S-Sets) and a single balancer class (M-Set) might suffice. The reviewer’s commentary on this particular point, *in extenso*, is:

The major problem of the manuscript is the classification of 5 classes of adaptation networks. The title is very attractive, but when I look through the main text and the supplement it is hard to find how the classification standard is chosen and why the way of classification is reasonable. Because this is the central point of the manuscript I would expect at least one figure to list all the five classes of motifs and explain the key features. There is also no such figure. Although figure 1 listed four of them but I have to go back and forth several time to realize figure 4 is the other one. When we really look into the 5 classes I am not sure it’s the right way to categories all the networks. Generally, networks in figure one are all feedback loop controlled networks while figure 4 feedforward loops. For feedback controlled networks the author further divide them into 4 classes. I find this extremely confusion. First single-opposer classes could be special cases of multi-opposer classes. I think there should be at

most two classes for S-set. These many classes just confuse people. Second all the opposer sets should use integral control as the mechanism which makes the feedforward loop dispensable. This means only one class is absolutely necessary to represent all the networks in the S-set.

As we explain in our introductory remarks, we completely agree with the reviewer's assessment that five different modular classes are superfluous. After careful consideration and analysis, we do indeed find that we can indeed express the full solution space of RPA network topologies in terms of **just two** classes of basis modules, which we define and name – one class of “opposer” modules (orchestrated by the “opposition” mechanism, and for which we had previously, and needlessly, subdivided into four different categories), and one class of “balancer” modules (orchestrated by the “balancing” mechanism). In particular, we have concluded that the reviewer is entirely correct in suggesting that single opposers may be considered special cases of opposing sets. In addition, we have now discovered a precise way to describe “feedforward” opposition (where opposer nodes, acting in feedback formation, *also occur in a route* and thus contribute a transmissive, feedforward, function in the network) in terms of the interconnectivity of modules, rather than incorporating this into the module itself as a topological feature.

The centrepiece of this work is the identification of network elements that can truly be considered **basis elements**, along with a **general way to combine** those elements, so as to **span the complete solution space** to the RPA problem. Remarkably, grouping our basis modules into just two classes (rather than five), and incorporating the novel phenomenon of feedforward opposition into the interconnectivity of modules has tremendously simplified our communication of the general RPA solution. Our descriptions of both the modules themselves, and their interconnections, have now become vastly more streamlined and elegant, and present to the reader a more accessible view of the overarching structures governing robustness in all complex networks.

But the central conclusion of our work remains unchanged: **All networks that exhibit Robust Perfect Adaptation (RPA) to a persistent change in stimulus are decomposable into well-defined modules, of which there exist several well-defined classes** – two general classes, scalable through the way they are connected for any size network, rather than the five we originally proposed. This novel finding represents a paradigm shift in our understanding of the basic structures underlying the complex networks occurring in nature. As we explain in our revised manuscript, “... these networks may now be considered from the point of their unexpected simplicity In this sense, complex bionetworks are like snowflakes: while each is unique, all individual instances are alike in their essential structure.”

The reviewer goes on to raise another important issue:

But actually if there are feedforward loops and feedback loops co-existing in one network the adaptation could due to either feedback control (S-set control) or feedforward control (M-set control), based on different parameters. The authors seem to totally neglect the importance of control parameters. All in all, I am not convinced the conclusion in this manuscript is sound and helpful to the field although I think the way the authors approach the problem is valuable.

We thank the reviewer for this helpful feedback. Our work presents general constraints for the reaction kinetics of the special “computational nodes” within RPA networks – opposer nodes, balancer nodes and connector nodes – with most of the technical details given in SI Sections 5.1 (for opposers), 5.2 (for balancers) and 5.3 (for connectors). In each case, we present the general constraint, and then give a number of concrete examples of reaction forms that would satisfy those constraints. But the reviewer is correct to point out that we do not clearly demonstrate how these examples of suitable reaction forms may actually be obtained from well-established rate laws such as Michaelis-Menten kinetics (for example) via appropriate parameter regimes.

In our revised Supplementary Information document, we now carefully elaborate on how each of our example rate laws may be closely approximated using Michaelian kinetics for enzymes that are either close to saturation (with very small Michaelis constants) or far from saturation (with very large Michaelis constants), depending on the requirements of the computational node in question.

In the case of opposer nodes, for instance,

Thus, to satisfy the opposer kinetics at (all) steady-states, π_n , the ratio of the positive and negative contributions to the overall reaction rate must be independent of P_o . This in turn requires that the functional form of the reaction rate, f_o , be separable in P_o , such that

$$f_o = h_o(P_o)g_o^+(P_u) - h_o(P_o)g_o^-(P_d). \quad (5.6)$$

In other words, the P_o -dependencies on the forward (f_o^+) and reverse (f_o^-) sides of the f_o equation must be commensurable. This principle imposes strict constraints on the types of reaction kinetics at P_o that can enable the node to function as an opposer.

Examples of appropriate reaction kinetics for an opposer node, could be, for example:

$$f_o = k_1 P_u - k_2, \quad (5.7)$$

or

$$f_o = k_1 - k_2 P_d, \quad (5.8)$$

or

$$f_o = k_1 P_u P_o - k_2 P_o, \quad (5.9)$$

or

$$f_o = k_1 (P_{Otot} - P_o) - k_2 P_d (P_{Otot} - P_o). \quad (5.10)$$

Equations (5.7) and (5.8) represent zero-order (in the substrate, P_o) regulations; Equation (5.9) encodes positive autoregulation by P_o , while Equation (5.10) encodes autoregulation of the reverse reaction by the inactive form ($P_{Otot} - P_o$), with P_{Otot} representing the (fixed) sum of the active (P_o) and inactive forms - a mass conservation constraint.

But how might these reaction forms actually be implemented by well-established rate laws for chemical reactions? Using Michaelis-Menten reaction kinetics as an example, Equations (5.7) to (5.10) could be *approximated*, respectively, by:

$$f_o = \frac{k_1 P_u (P_{Otot} - P_o)}{K_{m1} + (P_{Otot} - P_o)} - \frac{k_2 P_o}{K_{m2} + P_o}, \quad (5.11)$$

subject to the parameter constraints $K_{m1} \ll (P_{Otot} - P_o)$ and $K_{m2} \ll P_o$;

$$f_o = \frac{k_1 (P_{Otot} - P_o)}{K_{m1} + (P_{Otot} - P_o)} - \frac{k_2 P_d P_o}{K_{m2} + P_o}, \quad (5.12)$$

subject, likewise, to the parameter constraints $K_{m1} \ll (P_{Otot} - P_o)$ and $K_{m2} \ll P_o$;

$$f_o = \frac{k_1 P_u P_o (P_{Otot} - P_o)}{K_{m1} + (P_{Otot} - P_o)} - \frac{\hat{k}_2 P_o}{K_{m2} + P_o}, \quad (5.13)$$

subject to the parameter constraints $K_{m1} \ll (P_{Otot} - P_o)$ and $K_{m2} \gg P_o$, and where $\hat{k}_2 = k_2 K_{m2}$; and

$$f_o = \frac{\hat{k}_1 (P_{Otot} - P_o)}{K_{m1} + (P_{Otot} - P_o)} - \frac{k_2 P_d P_o (P_{Otot} - P_o)}{K_{m2} + P_o}, \quad (5.14)$$

subject to the parameter constraints $K_{m1} \gg (P_{Otot} - P_o)$ and $K_{m2} \ll P_o$, and where $\hat{k}_1 = k_1 K_{m1}$. In all these scenarios, a parameter constraint of the form $K_{mi} \ll P_i$, where P_i is the substrate in the associated enzyme-catalyzed reaction, corresponds to the enzyme being *saturated*, or close to saturation. A parameter constraint of the form $K_{mi} \gg P_i$, on the other hand, corresponds to the enzyme being *far from saturation*.

We thereby see, and emphasize in the revised manuscript, that although RPA does not require any specific parameter values (no fine-tuning of parameters needed), the correct computational function of opposer nodes may require effective Michaelis constants that reside in certain regions of parameter space.

Likewise, for balancer nodes, we provide some additional clarifications on the parameter regions that may be required for these nodes to execute their computational functions –

By way of simple example, consider a balancer node, P_B , with a single upregulator, P_u . The most general form for reaction rate at this node would be given by the relationship

$$f_B = f_B^+(P_u, P_B) - f_B^-(P_B),$$

which, in order for P_B to function as a balancer node, would have to assume the form

$$f_B = f(k_1 P_u + k_2)g(P_B) - f(k_3 P_B + k_4)g(P_B).$$

An example of appropriate reaction kinetics could therefore be

$$f_B = k_1 P_u - k_2 P_B. \quad (5.17)$$

Once again, we consider the question of how such a special form could be implemented, or at least closely approximated, by established rate laws for chemical reactions. Again appealing to Michaelis-Menten reaction kinetics, Equation (5.17) could be approximated by

$$f_B = \frac{k_1 P_u (P_{B_{tot}} - P_B)}{K_{m1} + (P_{B_{tot}} - P_B)} - \frac{\hat{k}_2 P_B}{K_{m2} + P_B}, \quad (5.18)$$

where $P_{B_{tot}}$ represents the (fixed) sum of the active and inactive forms of P_B , and subject to the parameter constraints $K_{m1} \ll (P_{B_{tot}} - P_0)$ (corresponding to the enzyme, P_u , operating *at saturation*, or *close to saturation*), and $K_{m2} \gg P_B$, (corresponding to the (constant) de-activating enzyme operating *far from saturation*) with $\hat{k}_2 = k_2 K_{m2}$. We see that the reaction kinetics for each of the set of balancer nodes associated to an M-set allow these balancer nodes to form a computational unit which constrains all balancer steady-states to a straight line trajectory parametrized by their associated D-node (see Theorem 5).

In Section 5.3, we explain that such parameter constraints do not obtain for connector nodes. Unlike opposers and balancers, whose reaction forms must be “paired” with respect to the substrates’ contributions to the activating and deactivating components of their reactions (thereby imposing the parameter restrictions we discuss above), connector kinetics are “paired” with respect to its regulating enzymes, thus obviating any constraints on suitable parameter regimes.

In closing, we would like to sincerely thank this reviewer again for such generous encouragement, and for many specific suggestions for improvement. We have thought carefully about all the reviewer’s comments, and gratefully acknowledge his/her contributions to the significant improvements we feel we have now made to our manuscript.

Response to Reviewer #2

We are deeply honored and truly appreciative of the huge investment of this reviewer's time, which has greatly helped us to communicate our arguments more simply, clearly and accurately, and to note the connections between our work and well-established results in engineering control theory. The reviewer has taken the special effort to write a very extensive document for our consideration which includes a brief "primer" on control theory. Since many comments and constructive criticisms of our paper are scattered throughout that document, even within the "primer" sections, we have decided to reproduce the reviewer's comments in their entirety below (*italicized and indented*) in order to ensure that we respond to each and every critique. In addition, we use **highlighted text to indicate the changes that we have made to our revised manuscript** in response to those criticisms.

Before giving detailed responses, point-by-point, to each of the reviewer's individual remarks, we would like to emphasize at the outset that the overarching goal of our work is to study the problem of Robust Perfect Adaptation (RPA), from a *topological* point of view, and in *full generality*, in complex networks that are self-organizing, self-regulating and evolvable – in other words, networks that arise in biology. We are not simply looking for *more* solutions to the RPA problem, given that we already know *some* solutions in very small, simple networks. Nor do we seek a method for *testing* a *particular* network topology to see if it satisfies the necessary mathematical conditions for RPA. Rather, we identify **for the very first time, the set of all possible RPA-capable network topologies** – not by enumeration, of course, or even by construction, but through the identification of a suitable **basis**.

As we explain in our newly revised manuscript, "**... In many biological contexts – cellular signal transduction and cellular metabolism, for instance – the underlying signaling networks are so complex and high-dimensional, so prone to change over time, and so extraordinarily variable from one realization to another (even from one cell to another phenotypically-identical neighboring cell), that the networks themselves are virtually impossible to define concretely at any useful level of detail.**" In view of the essential "unknowability" of most complex bionetworks, then, we propose to approach the problem of network complexity in reverse: Given an observed qualitative response (here, RPA), what conclusions can we draw with certainty as to the structure of the network responsible for that response? To work backwards from the observed response to the possibilities for the underlying network topologies, we need to determine **the full solution space** for the general RPA problem – that is, **ALL the possible networks that could exhibit RPA** in terms of the arrangement of nodes relative to one another, along with any constraints on the functions of those nodes (eg. via their reaction kinetics) within those arrangements.

The centerpiece of this work is the identification of a set of rich yet well-defined classes of network topologies which, together, span the space of all possible RPA-capable networks when suitably interconnected. These modular classes thus represent a **topological basis** for the solution to the RPA problem in any network, no matter how large. In this sense, the topological basis modules are like the "atoms" of robust adaptation.

Now, a major technical point raised by the reviewer is whether there are truly **five RPA basis modules** (four "opposer" modules, and a single "balancer" module), as we had claimed in

our original submission, or if there might really be just two generalizable modules. In our original submission, we had subdivided our “opposer” basis module into four distinct types based on two novel topological features – namely, the existence of “opposing sets”, where integral feedback control is transacted by a collection of nodes working together in collaboration within a feedback loop, and the existence of “*feedforward* opposers”, where nodes that participate integral *feedback* control also play a transmissive (*feedforward*) role in the network.

We have considered this matter very carefully, and do indeed find that our original subdivision of the opposition mechanism into four different basis modules was unnecessary and a distraction from the central results of our paper. In particular, we have concluded that a single opposer node may indeed be considered a special case of an opposing set (the “trivial opposing set”). In addition, we have now discovered a precise way to describe the phenomenon of “*feedforward* opposition” in terms of the interconnectivity of modules, rather than incorporating this into the module itself as a topological feature. Thus, as we explain in greater detail in our point-by-point responses below (and in our revised manuscript) we now conclude that the full set of network topologies that are capable of solving the RPA problem can be specified fully in terms of just two basis modules – one for the opposition mechanism (rather than four), and one for the balancing mechanism (as we reported previously).

Remarkably, grouping our basis modules into just two classes (rather than five), and incorporating the novel phenomenon of feedforward opposition into the interconnectivity of modules has tremendously simplified our communication of the general RPA solution. Our descriptions of both the modules themselves, as well as their interconnections, have now become vastly more streamlined and elegant, and present to the reader a more accessible view of the overarching structures governing robustness in complex networks.

Nevertheless, our central result still stands: **All networks that exhibit Robust Perfect Adaptation (RPA) to a persistent change in stimulus are decomposable into well-defined modules, of which there exist several well-defined classes** – two classes, as we now realize, rather than the five we originally proposed. This novel finding represents a paradigm shift in our understanding of the basic structures underlying the complex networks occurring in nature. As we explain in our revised manuscript, “... these networks may now be considered from the point of their unexpected simplicity In this sense, complex bionetworks are like snowflakes: while each is unique, all individual instances are alike in their essential structure.”

The second major technical query raised by the reviewer concerns the relationship of our present work to established results in control theory - in particular, the Internal Model Principle (IMP). We emphasize in this regard that our “RPA equation” is **not** intended to be a “centrepiece of this paper” (as the reviewer suggests), but a convenient algebraic form of the mathematical criterion for RPA which readily allows **a topological basis** for RPA-capable networks to be deduced (which **is** the centrepiece of our work). It certainly is true, as we now make more explicit, that our RPA equation is equivalent to the conditions given in Yi et al (albeit in an alternative form, with a single input and single output for reasons that have to do with the topological nature of the solution we seek). The essential difference

between our RPA equation, and the “IFC det” condition used in the Yi et al paper (and discussed by the reviewer), is one of form: with a single input and single output, there arises an unambiguous notion of “route” relative to the input/output pair, which is key to the delineation of the overarching topological structures of the network. In this way, our RPA equation makes explicit reference to network elements that are transmissive in nature (“routes”, which transmit biochemical signals from input to output) and elements that are regulatory in nature (“cycles” – encompassing reaction kinetics at each node (kinetic multipliers), as well as feedback loops (circuits)). As such, our RPA equation is particularly well suited to **identifying all the topological realizations of the IMP for RPA in complex, self-organizing, self-regulating networks.**

Of course, once we have a *particular* network topology in mind, we can readily test its potential to exhibit RPA *purely algebraically* – either using the IFC Det condition, or using the matrix form of the RPA equation $\det(M_{IO})=0$. But we respectfully emphasize that it is simply not the purpose of our work to produce a framework for testing special cases of putative RPA networks, or to explore well-established RPA instances such as the Che-gene signal transduction pathway involved in bacterial chemotaxis (first established as an RPA network by Barkai and Leibler in a seminal contribution to Nature in 1997, and later shown by Yi et al to be an instance of integral feedback control). The reviewer is right to point out that such an algebraic condition for the RPA problem has long been established.

We do make the connections between our work and previous results on the internal model principle more precise in our revised submission. For example, under the heading “Development of a General Analytical Framework”, we now explain that -

“In the 1970s, Francis and Wonham (21) investigated the necessary controller structures required to achieve robust regulation with internal stability, and established what is now referred to as the internal model principle (IMP). By this principle, a controller can reject exogenous disturbances and/or track prescribed reference signals by incorporating within itself a model of the dynamic structure of the disturbances/references. More recently, Yi et al (22) considered a special case of the IMP concerning RPA to constant exogenous inputs, in the context of providing a framework for understanding the extraordinary precision of adaptation in bacterial chemotaxis (5). This analysis provided a purely algebraic condition that must be satisfied by an RPA-capable system, which the authors showed to be equivalent to the requirement for integral control.”

We go on to explain that

“Here, our interest is not in confirming whether a particular network topology is capable of RPA, but in specifying all the possible network topologies – ie. all the possible arrangements of nodes that are capable of exhibiting RPA, along with any constraints on the reaction kinetics for those nodes. For this, we begin by developing an alternative version of the algebraic condition specified by Yi et al (22) for the special case in which a particular input/output node pair is specified, thereby constructing a framework from which topological structures may be deduced relative to that input/output node pair.”

On the other hand, we do emphasize in our newly revised Introduction section that

“... it is essential to recognise that biological systems differ in fundamental ways from engineering control systems. Molecular signaling networks are self-organising, self-regulating and evolvable, and as such, are comprised of elements that must serve both as the transmitted signals and their own controllers. Unlike their designed counterparts in engineering control systems, bionetworks do not have the luxury of employing specially-designed, dedicated components whose purpose is to sense or control biochemical signals. Although asymptotic tracking problems (of which RPA to constant exogenous inputs is a special case) have been studied extensively for engineering systems (21), how can we understand the mechanisms governing robust performance in the context of the self-organizing, self-regulating autonomous systems arising in biology?”

In particular, how can RPA-generating mechanisms be realized topologically in bionetworks – that is, in terms of the arrangements of interconnected nodes, along with the types of chemical reactions regulated by the interacting nodes? Do the topological requirements for RPA in large systems increase in complexity, or change qualitatively, along with the growth of the system, or do larger systems simply replicate the same basic design principles used by smaller systems? If the latter is the case, what are the universal topological principles that characterize general RPA-capable network designs (18)?”

In addition, we have created eight new figures for our extensively revised resubmission to clearly support our careful arguments as to the full set of all possible RPA network topologies through the identification of a *topological basis*. Of these, Figures 2 and 3 highlight the two different types of integral control that are realized, respectively, by the two different types of topological basis modules. We comment on these technical issues in greater detail in our point-by-point responses to follow.

By way of additional background on the scientific advance represented by our work, we gratefully acknowledge the reviewer’s constructive arguments towards the end of his/her review, and recognise that one can indeed construct more (larger) RPA networks (that satisfy the IFC Det condition) once one has identified a particular RPA network. But we respectfully point out that one simply cannot precisely delineate all possible topological arrangements of nodes that could solve the RPA problem by such descriptions of constructive methods. In fact, the reviewer’s constructive argument is reminiscent of an approach we ourselves had attempted in the early days of working on this problem. For small networks, it is relatively easy to find (at least some) ways to modify, or add to, simple known RPA solutions to obtain further RPA networks. For RPA networks using only the opposition mechanism, for example, this is a question of putting zeros in the right places in the matrix version of the RPA equation so that the matrix is always singular regardless of the values of the non-zero elements. For small networks, one can generally see by inspection when the RPA equation is satisfied due to the presence of these zeros: the form of our RPA equation gives rise to either a row or column of zeros in the simplest of cases. Even for slightly larger networks, one can permute the ordering of the nodes to obtain a form of the matrix that is readily observed to be singular by inspection due to the presence of the zeros. Likewise, for a pure balancing mechanism, singularity of the matrix comes from the linear dependence of collections of columns (or equivalently rows), and larger RPA networks using

only the balancing mechanism to achieve RPA can be certainly be constructed from smaller ones by preserving this property.

The difficulties of obtaining truly general solutions (topologically speaking, for *arbitrarily large and interconnected* networks) become more evident when one attempts to “**mix**” the two different mechanisms in sufficiently general ways as to obtain **all** solutions for a matrix (network) of a specified size. Once again, for fairly small networks, (some) constructions may readily be obtained that preserve the singularity of the matrix. As network size grows, some solutions may still be obtained and identified by inspection (here again, node ordering may be permuted to organise the matrix into blocks, with some blocks containing one or more rows or columns of zeros corresponding to the presence of the opposition mechanism, and other blocks containing linearly dependent row/columns corresponding to the presence of the balancer mechanism).

Not only is it impossible to organise these procedures into a truly general description of all the possible network topologies for any sized network (containing thousands of nodes, or even beyond), but more importantly, it is impossible to be sure when, if ever, one has indeed found all the topological realizations of RPA network of a given size. For this, we require a *topological* approach to the solution – an *abstract* consideration of the contents of the determinant expansion, with the topological roles of all the factors in each term duly noted, along with a general way to view partitions of that expansion into what we have termed “independently adapting subsets”. In doing so, we discover and characterise the topological basis sets of the RPA problem, which we show to correspond to RPA basis modules in the network. In our extensively revised manuscript, we now recognize that there are **two and only two such topological basis modules**. **As we take great pains to demonstrate rigorously, a network can ONLY exhibit RPA if it is decomposable into some combination of these modules**.

Several times throughout his/her review, the reviewer suggests that this novel **topological framework** for identifying the RPA basis modules, along with a set of “rules” as to how these modules may coexist (be interconnected) in larger networks, is “unnecessarily complicated”. We underscore in the pages to follow, and in the extensive revisions to our manuscript, that the task of identifying a suitable basis, along with a general way to combine those basis elements, to thereby yield the entire solution space for all possible RPA network topologies is indeed, inevitably, ***difficult*** – “complicated”, even – but not “unnecessarily” so. Among other things, we will attempt to explain how the constructive methods used by the reviewer readily lend themselves to omitting possible RPA solutions, particularly for networks that involve combinations of the two different types of modules.

We now proceed to a detailed point-by-point response to the reviewer’s criticisms below:

Reviewer #2 (Remarks to the Author):

I'm sympathetic to the aims of this paper but it needs major revisions to achieve them.

Details of review for NCOMMS-17-05100

This paper aims to expand and refine the concept of Robust Perfect Adaptation (RPA) as it arises in cell biology. The aims are interesting and even important, but I think the paper as written falls short. However, if the details were corrected and improved, the author’s aims

would be strengthened. So this review is critical of the technical details, but I'm sympathetic to the aims and would like to see a paper like this succeed. I'll sketch what the main aims are and then go into selected details that I think are the best opportunities for improvement. As I'm sympathetic to the aims of the paper, and the authors have clearly done a lot of work, I spent a lot (too much in retrospect) time going through the details in the supplement for anything that would justify the proliferation of definitions and mitigate the main problems focused on below. Unfortunately, I was unsuccessful, but the supplement is so long and involved it is hard to be certain.

Once again, we wish to express our sincere gratitude to this reviewer for such generous support of our work, and for investing so much time into working through our extensive Supplementary Information document which contains most of the technical details of our approach. We are very happy that the reviewer is “sympathetic to the aims of the paper” and “would like to see a paper like this succeed”. The reviewer’s extensive input has helped us to clarify our arguments and streamline our explanations of the general solution, topologically speaking, to the RPA problem in complex self-organizing/self-regulating networks.

While we acknowledge that our Supplementary Information document is indeed quite lengthy and involved, we respectfully point out once again that the overarching goal of our work is to identify **all possible topological realizations** of the two basic mechanisms of RPA (opposition and balancing) and their combinations, rather than to simply identify the two mechanisms themselves – an undertaking which demands a great deal of abstraction and mathematical rigor. Our topological focus, at such a complete level of generality, has required us to develop a new vocabulary (“opposition”, “opposing sets”, “balancing”, “kinetic multipliers”, etc) for speaking about RPA network topologies more abstract ways. The technical nature of the Supplementary Information document will inevitably not be of interest to all readers, but is nevertheless an important reference for those readers who seek a more complete mathematical understanding of the topological basis of the solution to the RPA problem. We have tried to streamline our explanations as much as possible in this newly revised submission, which has been greatly aided by the recognition that two classes of modules (rather than five) represent a complete topological basis for general RPA networks.

The reviewer continues:

The highlights:

Necessary conditions for RPA:

“... in stark contrast to previous work on the RPA problem (see (3,18) for recent reviews), we present a global and ‘top-down’ methodology that allows us to identify the necessary conditions for achieving RPA in arbitrarily large and complex networks.” (see line 57) The “previous work” cited is all recent papers in biology, where the literature is admittedly a bit of a mess. What is overlooked is a much bigger and older (>40 years) literature in

engineering and math in control theory. One centerpiece of this paper, the “RPA equation” is actually a special case of a special case of the Internal Model Principle (by Francis and Wonham, 1977). A very brief primer with this and additional reference is in -

Yi, Huang, Simon, Doyle: (2000) Robust perfect adaptation in bacterial chemotaxis through integral feedback control. Proc. Natl. Acad. Sci. 2000: Apr 25 97(9): 4649-53.

Yi et al appears to be the only biology paper like this, but it uses only the most trivial parts of control theory that was already decades old in 2000. The RPA equation is trivially equivalent to the RPA conditions given in Yi et al. I'll expand on this in more detail below.

As part of the revision process for our manuscript, we have carefully reviewed the work of Francis and Wonham, and Yi et al, and thank the reviewer very much for bringing this key literature to our attention. We do not deny that our RPA equation is equivalent to the conditions given in Yi et al (and is actually a special case tailored to our particular topological goals for this work). Our revised manuscript makes this clear, and highlights what our work accomplishes that previous studies (such as Yi et al.) do not: that is, we show – in complete generality – how the necessary conditions for RPA (to a constant exogenous input) are realized **topologically** in self-organizing, self-regulating bionetworks of arbitrary size and complexity.

We comment on these distinctions in greater detail later in our response, since the reviewer raises variations on this point several times. But for now, we reiterate: Our approach identifies not only the “mechanisms” for RPA (ie. requiring integral control, of which there are two different types), but *their topological implications* – how signaling nodes must be arranged relative to each other, and (as we show) organized into well-defined modules which may be connected together (subject to a well-defined set of intermodular connection rules). Our major innovative leap, in comparison with previous work, is to show – rigorously and conclusively - that the topology of **any** RPA-capable network can be expressed as a combination (interconnection) of modules, selected from the two classes of topological basis modules.

The reviewer continues:

Five and only five classes

Beyond the RPA equation, a central claim in this paper is that (line 86) “...five and only five distinct classes of network modules represent a basis for the solution to the RPA problem in any size network.” The aim here is interesting but the claim appears not to be correct. The 5 classes are in fact only 2 in disguise, though the disguises are quite clever. I'll work through the details of this below, and the required machinery is vastly simpler than what is in the paper.

The reviewer is entirely correct to suggest that only two classes of modules are needed to solve the general RPA problem, and we have extensively revised our work to reflect this. Our work demonstrates conclusively that there are *two and only two mechanisms* for generating RPA (which we refer to as opposition and balancing), and we now show that each mechanism engenders a distinct class of topological basis modules. We clarify the nature of these topological basis modules much more carefully in our revised manuscript (and Supplement), and also explain more carefully the constraints as to how these modules may be interconnected to form larger (multi-modular) networks. We comment more extensively on the nature of these intermodular connectivity constraints later in our response.

The reviewer continues:

Minimal modules

The authors also seem to have overlooked a crucial minimal solution in their list of “minimal modules” in S6.2 (page 47). Below I’ll describe what I find to be the two “fundamental” motifs. This has all been well-known in engineering for decades, but is no longer emphasized and would deserve some revival in the context of biology. The paper would be much clearer if such minimal modules were corrected and emphasized more rather than devote many pages to algebra for what seem to be unnecessarily complex case studies.

It is true that we did not include the minimal version of the balancer module in our original list of “minimal solutions”. Although we have fundamentally re-organized our presentation of the general topological solution to the RPA problem, we point out here by way of explanation that the original depictions of minimal solutions were intended to be *minimal networks* that incorporated what we previously identified as five basis modules. In that earlier version of our solution, we had incorporated, for instance, the topological possibility of an opposer node (which “opposes” routes while in *feedback* formation to those routes) that *is also in a route* (a “feedforward opposer”, in our previous terminology) within the description of the module itself (a “feedforward opposer module”).

But again, we emphasize that we no longer use this five-module categorization to describe the full (topological) solution space to the RPA problem. We agree with the reviewer that two classes of modules are sufficient, and we have reorganized our explanations accordingly. In our new Figure 8, for example, we present illustrations of “small RPA networks with new topological features”, and explain the smallest networks that can incorporate, among other things, two *different* types of RPA module working together in collaboration to allow the network as a whole to exhibit the RPA property. We return to this matter later in our response, as the reviewer raises variations on this issue several times.

Next, the reviewer offers us a primer on integral feedback control theory basics, which we gratefully acknowledge:

Primer on integral feedback control theory basics: *The paper mentions integral feedback control (IFC) but never clearly states what it is, though it is implicit in much of the supplement. Since IFC is such a simple concept it is probably worth explicitly mentioning what it is. The RPA equation, a centerpiece of this paper, is a simple special case of the IFC det condition below, which is a special case of the Internal Model Principle (IMP). The paper incorrectly claims that the RPA equations is a “generalization of integral control” (191) when it is a special case of a special case of a result (IMP) that is now >40 years old. For references on IMP and a brief primer on Integral Feedback Control (IFC) for biologists, see Yi et al above.*

I’ll quickly review below what is in Yi et al 2000 PNAS paper but note it has been “well known” (taught to undergrads) regarding “perfect adaptation” in engineering for at least 30 years. The current submission’s main results are largely special cases, so the paper would be both more scholarly and clearer if it adopted these standard results. The confusion arises on line 57 where the paper states that “...until now, three basic approaches have been used to understand the requirements for RPA (2,3,5,8,14,18,19), and they have only provided answers for very small systems.” The authors claim that “... in stark contrast to previous work on the RPA problem (see (3,18) for recent reviews), we present a global and ‘top-down’ methodology that allows us to identify the necessary conditions for achieving RPA in arbitrarily large and complex networks.” If you ignore the Yi et al PNAS paper above, this is largely true in the biology literature but has emphatically not been true for >40 years in undergrad engineering control theory. What follows is a minimal primer on this material.

We are truly appreciative of the reviewer’s extensive input on the relationship of our work to established engineering control theory, and suggestions as to how the concept of integral feedback control could be referenced more explicitly in our work.

The reviewer points out at this stage that the RPA equation is “a centerpiece of this paper”, and we would like to acknowledge here once again that our RPA equation is, indeed, as the reviewer correctly points out, an alternative statement of the “IFC Det condition”. We elaborate on our decision to use *this particular form* as our RPA equation in more detail later in the context of why this particular incarnation of the principle is best suited to our (topological) purposes. For now, we simply wish to emphasize again that the RPA equation is not intended to be a “centerpiece” of this work; rather it is the launching point for our topological solution methodology concerning the partition of the terms of the RPA equation into basis subsets (“independently adapting subsets”). It is our method for solving this equation in a way that can view **any** RPA-capable network in terms of interconnections of RPA basis modules that is intended to be the centerpiece of this work.

In addition, we would like to respectfully point out that we do **not** claim that “*the RPA equations is a “generalization of integral control” (191)*”, as the reviewer suggests. What we had actually stated in that section of our original submission was that “the **opposition mechanism** represents a generalization of **integral feedback control** to **networks** of arbitrary size and complexity” (emphasis added). Recall, in this regard, that our work identifies two *different* RPA-generating mechanisms – opposition and balancing – along with their topological realizations (opposer modules and balancer modules). The balancing mechanism underlies a different form of integral control that does **not require feedback**,

but requires **multiple “routes”** (and hence a “feedforward” topology), **and** requires the associated module to adhere to its “linearization” at steady-state for all inputs. To be fair, we now notice that we had unwittingly omitted the key word “feedback” from this particular sentence in our original submission (which was certainly an error on our part). We also realize that we should, more correctly, have stated that opposer *modules* represent *topological* generalizations of *integral feedback* control in self-organizing/self-regulating networks of arbitrary size and complexity.

Based on the reviewer’s extensive and constructive comments, we do recognize that our original presentation of these technicalities and distinctions was inadequate. We have now clarified the relationship of our study to “well-known” results in control theory (concerning the internal model principle). Under the section heading “Development of a General Analytical Framework”, for example, we explain that –

“In the 1970s, Francis and Wonham (21) investigated the necessary controller structures required to achieve robust regulation with internal stability, and established what is now referred to as the internal model principle (IMP). By this principle, a controller can reject exogenous disturbances and/or track prescribed reference signals by incorporating within itself a model of the dynamic structure of the disturbances/references. More recently, Yi et al (22) considered a special case of the IMP concerning RPA to constant exogenous inputs, in the context of providing a framework for understanding the extraordinary precision of adaptation in bacterial chemotaxis (5). This analysis provided a purely algebraic condition that must be satisfied by an RPA-capable system, which the authors showed to be equivalent to the requirement for integral control.

“Here, our interest is not in confirming whether a particular network topology is capable of RPA, but in specifying *all the possible network topologies* – ie. all the possible arrangements of nodes that are capable of exhibiting RPA, along with any constraints on the reaction kinetics for those nodes. For this, we begin by developing an alternative version of the algebraic condition specified by Yi et al (22) for the special case in which a particular input/output node pair is specified, thereby constructing a framework from which topological structures may be deduced relative to that input/output node pair.

“The generality of our method for studying RPA in biological networks, being self-organizing, complex and evolvable, builds upon precise definitions of all the key terms of the problem, which we provide in detail in the attached Supplementary Information (SI) ...”

But we also emphasize earlier on, in the Introduction section, that –

“ ... it is essential to recognise that biological systems differ in fundamental ways from engineering control systems. Molecular signaling networks are self-organising, self-

regulating and evolvable, and as such, are comprised of elements that must serve both as the transmitted signals *and their own controllers*. Unlike their designed counterparts in engineering control systems, bionetworks do not have the luxury of employing specially-designed, dedicated components whose purpose is to sense or control biochemical signals. Although asymptotic tracking problems (of which RPA to constant exogenous inputs is a special case) have been studied extensively for engineering systems (21), how can we understand the mechanisms governing robust performance in the context of the self-organizing, self-regulating autonomous systems arising in biology?

“In particular, how can RPA-generating mechanisms be realized *topologically* in bionetworks – that is, in terms of the arrangements of interconnected nodes, along with the types of chemical reactions regulated by the interacting nodes? Do the topological requirements for RPA in large systems increase in complexity, or change qualitatively, along with the growth of the system, or do larger systems simply replicate the same basic design principles used by smaller systems? If the latter is the case, what are the universal topological principles that characterize general RPA-capable network designs (18)?”

Moreover, on the subject of the computation of integrals by the opposition mechanism,

“... we illustrate in Figures 2a and 3 (with additional discussion in SI Section 5.1) that opposer nodes play a special computational role in RPA networks by calculating an integral of a “tracking error” – the difference between some network quantity (eg. the activity of a particular node) and its steady-state value, the latter being determined purely by system parameters, rather than the magnitude of the system input.

“For a single opposer node, the tracking error in question corresponds to the error in the activity of the single independent regulator (Figure 2a). Since the opposer and its regulator participate in a common circuit, the computation of this integral constrains the regulator node to exhibit the RPA property. In an opposing set, on the other hand, each opposer in the collection computes the integral of a unique tracking error, involving various combinations of nodes in the master set (Figure 3). All nodes featuring in these various tracking errors exhibit the RPA property due to the combined effect of the multiple integrals; the opposer nodes themselves, by contrast, never exhibit the RPA property (SI Section 5.1). In any event, all these computations work together to confer the RPA property on one or more nodes *within the route being opposed*. We shall see later that the fact that some nodes within the opposer module exhibit the RPA property, while others do not, has important implications for the possible interconnectivities of modules in larger networks.

“In any case, by distributing the requisite integral over multiple opposer nodes in this way, opposing sets represent a topological generalization of the application of integral feedback control to complex, self-organizing, self-regulating bionetworks.

“This novel concept of an opposing set, where multiple nodes work together in complex interlinked feedback relationships to compute RPA-conferring integrals, has never previously been reported to our knowledge. Nevertheless, it is apparent that these more complex arrangements of opposer nodes are indeed employed in certain gene regulatory circuitries since we show in SI Section 6.1.5 that the recently-identified phenomenon of antithetical integral feedback (3,14) is actually an instance of a two-node feedback opposing set with single input/output node.”

On the subject of opposing sets, we offer the following clarifications on their relationship to single opposer nodes -

“The topological requirements of opposing sets specified in Theorem 3, combined with the requirement for each opposer node to have a single independent regulator in a common circuit, define a rich class of network topologies associated with the opposition mechanism: a collection of opposer nodes distributed to a set of interlinked circuits, embedded into a feedback loop that is contiguous with the route being (fully) opposed. In this sense, a single opposer (with no disjoint circuits relative to a route it fully opposes, and is embedded alone into a contiguous circuit), may be considered a *trivial* opposing set – a special case which vacuously satisfies the conditions of Theorem 3.

“In Figure 1(a,b), we depict the class of network topologies corresponding to the mechanism of an opposing set, illustrating (for the sake of definiteness) the case of a single opposer node as well as a simple version of a two-node opposing set. We refer to these network topologies hereafter as *opposer modules*. Some additional examples of opposing sets are depicted in Figure 3, with a more general representation of opposing sets given in Figure 4 (further details in SI Section 5).”

In addition, we acknowledge the correctness of the reviewer’s later arguments that the integrator “state” for an opposing set is a linear combination of states. Nevertheless, we have important reasons for wishing to depict opposing sets as collections of individual integrators, working together in collaboration, for reasons that have to do with allowed intermodular connections for opposer modules that employ an opposing set. We comment on our presentation of the essential ideas of integral control for the two types of module in greater detail later in our response.

The reviewer continues:

Starting simply, consider the linear system

$$[x'; y] = [A \ b; c \ d][x; w] = [A \ b; c \ 0][x; w].$$

where y and w are scalar output and input, and x is a vector internal state and all vectors and matrices are real and of compatible dimensions. It is actually simpler to use this general case rather than the special case where some states are identified as inputs and/or outputs. We'll assume that $c \neq 0$ for nontriviality (and we are assuming $d=0$ as in the paper, not an important restriction).

The reviewer claims here that “*It is actually simpler to use this general case rather than the special case where some states are identified as inputs and/or outputs*”. In the context of what we are actually attempting to achieve in this study – namely, to define the **full** solution space of all network **topologies** for complex networks that are self-organizing, self-regulating and evolvable, whose nodes *represent both the signal and their own controllers* – we respectfully disagree with the reviewer’s point of view on this matter. We sincerely thank the reviewer for helping us to communicate more clearly how our work is distinctly different in its approach, and in its end goal, from previous work. After carefully reviewing the two early papers by Francis and Wonham, and also that by Yi et al, we certainly recognize that the form of the linear control problem presented by the reviewer (and the corresponding form of the “IFC Det criterion” referenced by both the reviewer and Yi et al), is considerably more general and considers (among other things) input and output “states” (so to speak) that are effectively linear combinations of various individual component states. We maintain, however, that it is highly convenient to our purpose to select a single input node (the node which receives a stimulus or perturbation delivered from outside the system), and a single output node (the network node that represents the endpoint of interest). Since our goal is to identify **all** network **topologies** that are capable of solving the RPA problem, a particular choice of input/output node pair produces an unambiguous notion of “route” – ie. a “pathway” of biochemical signal transmission through the network (ie. from input to output). As we explain later, this clear delineation of the specific topological roles played by all nodes of the network, and especially those nodes with a computational (integral computing) role, greatly aids us in drawing the strong conclusion as to the required modularity of RPA capable networks with respect to a chosen input/output node pair. Importantly, our particular interpretation of the algebraic criterion for RPA most closely corresponds to what we would actually *measure* experimentally: complex networks would typically be perturbed, or stimulated, via the delivery of a particular growth factor, say, or a target-specific enzyme inhibitor, or perhaps by knocking-down a specific gene product by siRNA; the effect of the perturbation on various individual endpoints of interest would then be studied.

The reviewer goes on:

We'll consider some arbitrary initial condition in x and a step input in w , both at $t=0$. Then “perfect adaptation” (PA) is defined to be the property that

$$PA: \quad y(t) \rightarrow 0 \quad \forall x(0) \text{ initial conditions} \quad \forall w(t) = \text{constant (step)} \\ t \rightarrow \infty$$

This goes by other names in engineering (asymptotic tracking, zero steady state, ...) but we'll use the biologist's PA. For Robust PA (RPA) we need to define some uncertain set $[A, b, c] \in S$ and check that PA holds for all $[A, b, c] \in S$. We'll assume A is stable throughout. Stability is largely ignored in this paper, and discussed briefly in S9 (Supplement Section 9). More on this later, but note that for any nontrivial uncertainty model, verifying robustness of PA is trivial compared with robustness of stability, and thus not surprisingly robust control theory focuses on the latter. In biology the relative simplicity of testing RPA should probably be a feature.

The main “integral feedback control” (IFC) theorem is

$$IFC \text{ Thm: } PA \text{ iff } \det [A \ b; \ c \ d] = 0 \Leftrightarrow \exists k \text{ s.t. } z = k \ \& \ z' = y.$$

This is easily proven with a few lines of algebra, given in the Yi et al paper. This is arguably the simplest nontrivial result in control theory and is a special case of the Internal Model Principle (IMP), that PA to an external disturbance like a step requires an internal model. For example, PA to steps, ramps, and sinusoids need, respectively, an internal single and double integrator, and an oscillator. Let me underscore here that the “RPA equation” is strictly a special case of the det condition in the IFC Thm (easily shown by simply taking the special case in the paper and applying the Schur complement formula to the determinant). In the author's defense these simple results are perhaps not adequately emphasized in control theory education, and typically mentioned mostly in passing, as RPA seems to be such a trivial special case that it doesn't need highlighting. I think this attitude makes sense in engineering but perhaps less so in biology. In any case, biologists are clearly interested in RPA so it would help them to have a rigorous and accessible account.

Once again, we completely agree with the reviewer's statement that the “IFC Theorem”, and consequently our RPA equation also, is “a special case of the Internal Model Principle (IMP), that PA to an external disturbance like a step requires an internal model”. We completely concur that “the 'RPA equation' is strictly a special case of the det condition in the IFC Thm (easily shown by simply taking the special case in the paper and applying the Schur complement formula to the determinant)”. On the other hand, the reviewer claims that “RPA seems to be such a trivial special case that it doesn't need highlighting. I think this attitude makes sense in engineering but perhaps less so in biology.” We feel it is worth pointing out that RPA (as we define it in this work) is of **fundamental importance** in biology. As we emphasized earlier, network perturbations in biology most commonly come in the form of the arrival (or upregulation/downregulation) of some signaling protein (eg. growth factor, mitogen, inflammatory cytokine, etc.), the mutation (or knockout) of a gene, or the delivery of a pharmacological inhibitor, for example. These perturbations are typically realized on a very short time scale in comparison with the regulation of the network, and are thus most readily approximated by a step function. RPA to other more general inputs such as ramps and sinusoids, are not of particular interest in this context. Once again, our overarching goal is to identify the *full solution space of network topologies* that are capable of exhibiting RPA to the kinds of perturbations we describe here – that is, exogenous inputs that are effectively *constants*.

We interpret RPA in this sense throughout our work. We explain that –

“Robust Perfect Adaptation (RPA) is the ability of a system to generate an output that returns to a fixed reference level (its “set point”) following a persistent change in input stimulus, with no need for tuning of system parameters (1-3). RPA has been widely observed throughout biology (1,4,5-11), at the cellular level (signal transduction, gene regulation, protein expression (6-8)), in sensory systems (6,10,11), at the whole-organism level in mammals (9), and during development (12,13). For example, mammalian plasma calcium concentration exhibits perfect adaptation to persistent changes in calcium export (e.g. lactation), or influx (e.g. diet changes or bone remodeling), thereby keeping plasma calcium within very tight tolerances as calcium demands vary (9). In addition, perfect adaptation enables a biological system to “reset” itself following a perturbation, in order to maintain responsiveness to subsequent variations in external stimuli (3). The RPA property thus promotes high sensor sensitivity, while increasing the dynamic range, regardless of the intensity, or the variations, in the average stimulus (1-3,14,15).

“Importantly, while RPA confers many benefits to living systems, loss of the RPA property in networks that require it could lead to disease (e.g. ras-mediated oncogenesis (3,16)), reduced fitness (1), or death (17).”

The reviewer continues:

The integral feedback is in the variable $z = kx$ which has dynamics $z' = y$. This is popular in engineering because it is easily implemented in digital controllers, and the main technical issue is called “integrator windup” which occurs with saturating actuators, and thus integral controllers include antiwindup mechanisms. In biology, the issue is more “reverse engineer” to identify mechanisms that achieve PA, and the paper cites several of the standard papers in this genre. The det condition is a simple algebraic test for PA and is equivalent to the existence of integral control.

Beyond the RPA equation, a central claim in this paper is that (86) “... fundamental solutions to the RPA problem is limited to five: that is, five and only five distinct classes of network modules represent a basis for the solution to the RPA problem in any size network.” I was unable to find a clear definition of what “fundamental solutions” are but it seems fairly clear to me that their five distinct classes are in fact elaborations of just two “fundamental” motifs.

We concur that there are indeed two, rather than five, “fundamental” motifs – by which we mean *topological basis modules*. These important technical details have all been carefully revised in our submission.

The reviewer then moves on to the topic of “minimal motifs” -

The authors also seem to have overlooked a crucial minimal solution in their list of “minimal modules” in S6.2 (page 47). I’ll next describe what I find to be the two “fundamental” motifs, the first of them equivalent to S6.2.1 Case A on page 49, but the second one is a nonlinear “balancer” module that is simpler than S6.2.3 Case C on page 49. This has all been well-known in engineering for decades, but is no longer emphasized and would deserve some revival in the context of biology.

The reviewer correctly points out that we omitted the minimal balancer module from our original listing of minimal modules. Our original intention in presenting a selection of “minimal networks” was to highlight how many nodes would be required for a network to exhibit RPA while including the novel topological features we had identified – in particular, “feedforward opposition” (where an opposer node, which operates in feedback formation to the route it opposes, *also* operates in a route), and opposing sets. In our previous framework, we had used these new features to make unnecessary topological distinctions among opposer modules. Again, we have discarded this approach in our revised submission, and agree with the reviewer that two modules are sufficient to describe the set of all possible RPA-capable network modules. As we explain in our new manuscript, and elsewhere in this document, the phenomenon of “feedforward” opposition is now incorporated into the interconnectivity of modules, rather than as a feature of the modules themselves, thus vastly streamlining our explanations of the complete RPA solution space as the arbitrary combination (via allowed interconnections) of RPA basis modules. Likewise, we now make clearer that single opposer nodes are, essentially, *trivial* opposing sets, thereby allowing us to dispense with the concept of making opposing sets into a separate type of module. Our new Figure 8 communicates more carefully our intended purpose for these “small network” examples - to highlight to the reader that we would need to solve vastly larger computational problems if we were to identify these kinds of RPA-capable topologies using “blind” methods such as computational screening. This is important because the study by Ma et al (that we cite several times in our paper), published in *Cell* in 2009, undertook a huge computational search of “the 16,038 possible three-node topologies that contain at least one direct or indirect causal link from the input node to the output node” (what we would call “connected and transmissive” networks containing three nodes), using Michaelis-Menten reaction kinetics, with 10,000 different parameter sets *for each individual topology*. They thereby analyzed a total of $(16038) \times (10000) \approx 1.6 \times 10^8$ different circuits. Their results showed that, for networks containing just three nodes, all RPA solutions are either a minimal version of what we call an opposer module (using a single opposer node), or a minimal version of what we call a balancer module. It is clear that searching for opposing sets, or combinations of opposer and balancer modules, for instance, using such methods, would be so computationally expensive as to be effectively infeasible.

For explanatory purposes, we wish to point out that our former “Case C” (now “Case a” in our new Figure 8) was intended to reflect an RPA-capable network that *included* a (minimal) opposer module, whose single opposer node *was also in a route* (hence, a “single feedforward opposer module” in our previous categorization of the modules). In that earlier framework, we emphasized that such modules require the collaboration of an additional module that either “balances” or “opposes” the route in which the “feedforward opposer” participates; in that particular example, we chose a (minimal) balancer module for this purpose. In other words, our former “Case C” represented a minimal RPA **network** that featured **both** an opposer module **and** a balancer module connected together. (Note that, in that example, if the opposer node (noted in yellow) failed to exhibit “opposer kinetics”, the output node O would not exhibit RPA in that network. Our former Case C was, most emphatically, not just a balancer module with some extra “unnecessary regulations”: again, this is an opposer module and a balancer module connected together “in series”. In the same vein, our former Case F (now “Case f” in our new Figure 8) is a two-node opposing set (involving nodes O_1 and O_2) connected “in series” with a single opposer module (involving O_3).) We give detailed explanations on the meaning of modules being connected “in parallel” and “in series” in our revised manuscript, and also later in this document.

We clarify the nature of these small network solutions much more carefully in our revised submission, and clearly explain to the reader what we are trying to show using these simple illustrations. (We note for reference that our former “Case C” has become “Case a” in our new Figure 8; our former “Case E” has become “Case d”; our former “Case F” has become “Case f”; our remaining examples, Cases b, c, and e are all new). We explain, for instance, that -

“In Figure 8, we consider the smallest RPA networks that are capable of invoking the various novel topological features we identify in the present work, illustrating the significant increases in the computational screening problems that would be required to identify these topologies. For a network to employ both an opposer module and a balancer module working together in collaboration, for instance, a minimum of five nodes would be required (Figure 8a,b). Likewise, for a network to feature an opposer node that is also involved in a route (Figure 8a,c), thereby requiring an additional ancillary module connected in series, at least five nodes are needed. For a network with distinct input/output nodes to incorporate a (non-trivial) opposing set (Figure 8d), five nodes are, once again, the minimum requirement. If one or both of these opposer nodes are also in a route, however, thereby requiring the collaboration of an ancillary module (Figure 8e,f), the smallest such RPA networks contain seven nodes.”

We will elaborate further on these small network illustrations in the sections to follow.

The reviewer continues -

Minimal “motifs”: Both minimal motifs have 2 states. The first is the pure example of integral feedback control (IFC) and can be used to illustrate a simple but meaningful notion of robustness. Consider the system

(GRAPHIC: case 1)

This also has a variety of robustness features. For example consider the same sparsity pattern (GRAPHIC: same sparsity pattern illustrated)

Note that the det condition (and stability) holds for all $a_{ij} > 0$ and the integrator is trivially in the 2nd state. This is the purest example of RPA and all but one of the “5 distinct classes” are simply elaborations of this where the pure integrator is replaced by a circuit that implements an integrator. More details on this later.

A second case which biologists call Incoherent Feedforward (IFF) has a linearization of (GRAPHIC: case 2)

but here the integrator is a mixture of the 2 states. This is not as robust as the first case, in that

(GRAPHIC: same sparsity pattern illustrated)

So that it appears that the parameters must be “fine-tuned” so that $a_{21} = b_2$. However, a nonlinear version can be easily made robust, and the authors seem to have rediscovered this, though made it unnecessarily complicated. Consider the nonlinear system with input w and output P (using notation similar to the paper) and additional state B

(GRAPHIC)

so the output P has RPA with respect to the parameter a . The linearization around $w=0$ is a version of Case 2:

(GRAPHIC)

So the linearization appears to need fine-tuning but since $a_{21} = b_2 = a$ this is done “for free” by the linearization of the nonlinear system. This has been viewed primarily as a curiosity in engineering, but it appears from earlier work in biology that this nonlinear motif does arise natural in biology and deserves further study. In the paper this case study is not identified as one of the “minimal” modules but is part of S6.2.3 Case A which has 3 additional (and unnecessary) states.

Again, we sincerely thank the reviewer for investing so much time into conveying to us these various ideas for consideration. Once again, on the subject of “minimal motifs”, we had originally intended to highlight how large computational screening problems would have to be in order to “discover” feedforward opposition, opposing sets, and compound networks (comprising multiple modules connected together – just two different modules, in the very simplest cases) by such methods. Now that we have revised our explanatory framework for the complete solution space to the RPA problem in terms of just two (rather than five) topological basis modules, our earlier presentation of these minimal networks is no longer relevant.

The reviewer’s “case 1” (equivalent to one of our original “minimal modules”) is a two-node RPA network comprising just one opposer module, generated by a *single* opposer node (which, as we now point out in our revised analysis, may be viewed as a “trivial” opposing set). Here the integrator is the opposer node itself. The “case 2”, on the other hand, is the

minimal version of a single balancer module (which we omitted from our original selection of minimal networks). As the reviewer points out, this module makes a very different type of computation from that of an opposer module. The reviewer correctly points out that the balancer module strictly requires the linearization of the nonlinear system in order to guarantee robustness (ie. no fine-tuning of parameters required).

Regarding the relationship between this linearization, and the robustness of perfect adaptation generated by balancer modules, we would like to make several additional points. It's essential to recognize that, although our RPA equation (and correspondingly the "IFC Det" condition that appears in Yi et al) is obtained from a linearization of the system about the steady-state, our analysis shows that *special classes of reaction kinetics* are required at what we denote the "balancer nodes" in order to constrain these nodes to adopt steady-states values that are linearly related to what we call the D-node (diverter node) . The D-node occurs at the apex of the module, and in the "minimal" balancer module discussed in this section by the reviewer, the module's D-node is simply the input node. We discuss these special classes of reaction kinetics in our Supplementary Information document, Section 5.2; in our revised version, we elaborate on how these special reaction forms could be implemented, or at least approximated, by established rate laws such as Michaelis-Menten kinetics, explaining that –

By way of simple example, consider a balancer node, P_B , with a single upregulator, P_u . The most general form for reaction rate at this node would be given by the relationship

$$f_B = f_B^+(P_u, P_B) - f_B^-(P_B),$$

which, in order for P_B to function as a balancer node, would have to assume the form

$$f_B = f(k_1 P_u + k_2)g(P_B) - f(k_3 P_B + k_4)g(P_B).$$

An example of appropriate reaction kinetics could therefore be

$$f_B = k_1 P_u - k_2 P_B. \quad (5.17)$$

Once again, we consider the question of how such a special form could be implemented, or at least closely approximated, by established rate laws for chemical reactions. Again appealing to Michaelis-Menten reaction kinetics, Equation (5.17) could be approximated by

$$f_B = \frac{k_1 P_u (P_{B_{tot}} - P_B)}{K_{m1} + (P_{B_{tot}} - P_B)} - \frac{\hat{k}_2 P_B}{K_{m2} + P_B}, \quad (5.18)$$

where $P_{B_{tot}}$ represents the (fixed) sum of the active and inactive forms of P_B , and subject to the parameter constraints $K_{m1} \ll (P_{B_{tot}} - P_0)$ (corresponding to the enzyme, P_u , operating at saturation, or close to saturation), and $K_{m2} \gg P_B$, (corresponding to the (constant) de-activating enzyme operating far from saturation) with $\hat{k}_2 = k_2 K_{m2}$. We see that the reaction kinetics for each of the set of balancer nodes associated to an M-set allow these balancer nodes to form a computational unit which constrains all balancer steady-states to a straight line trajectory parametrized by their associated D-node (see Theorem 5).

In other words, evolution has to "find" these special enzyme-substrate reaction kinetics at balancer nodes in order for the associated module to promote RPA in a nonlinear network.

As the above excerpt from our Supplementary Information document conveys, for Michaelian enzymes this could require operating either *at saturation* (in the case of the activating reaction above), or *far from saturation* (in the case of the de-activating reaction above). We refer to the special reaction kinetics required at balancer nodes as “balancer kinetics”. The entries in the matrix forms of the “IFC Det” condition, or the RPA equation, are not just numbers: they are “generated” by the reaction kinetics at the associated nodes.

In this way, the reaction kinetics of balancer nodes contribute a key computational function to RPA networks. We highlight this subtlety in our depiction of integral control for a “minimal” balancer module in Figure 2b. In that figure we emphasize that the balancer nodes are required to adopt steady-states that are in a strict linear relationship with the D-node (and with each other, in balancer modules that contain multiple balancer nodes) by expressing their activity in terms of an integral of a tracking error. This particular tracking error concerns the deviation of the node’s activity from the “flat manifold” to which it must adhere at steady-state. Once this “linearizing” integral has been computed at all balancer nodes, the connector node (C-node) is able to compute an integral which allows its steady-state to be independent of the activity of the D-node, provided the connector node has the requisite reaction kinetics to compute such an integral. We elaborate on the reaction kinetics required for connector nodes (“connector kinetics”) in SI Section 5.3. All these principles are clearly illustrated to the reader by way of the simple example presented in Figure 2b (with corresponding numerical simulations presented in Figure 2d). While the reviewer only identifies the C-node as the “integrator” for the module, our simple illustration emphasizes that integral control in balancer modules involves a computational collaboration with two different types of nodes - one or more balancer nodes, whose linearizing ability creates the conditions that allow the C-node to complete the “balancing act”.

We also point out that this “balancing” mechanism, while “*viewed primarily as a curiosity in engineering*” is by no means a “curiosity” in the robust performance of complex networks of chemical reactions. In fact, most examples of simple RPA networks identified to day involve this mechanism. Transcription networks in E Coli, for instance, being a particularly well studied model system, is replete with examples of three-node networks of this type. Fewer examples have been identified of the opposer type, although the simple feedback control circuit underpinning chemotaxis in E Coli identified by Barkai and Leibler (and later analyzed by Yi et al) is a well-known example.

There is really no reason to consider the opposition mechanism more important than the balancing mechanism in complex bionetworks, just because this is the more “natural” approach in engineering control systems: simple versions of both modules have been identified in bionetworks; both require “special” reaction kinetics at certain key nodes, and

we have identified in our study the classes of reaction kinetics that are required to support the essential computational functions of these nodes.

The reviewer continues:

While the authors seem to overlook the minimality of Case 2, they would agree with this analysis that Case 1 and 2 are distinct cases. Where we would appear to disagree is that I can easily show that all of their 5 “distinct cases” are in fact nearly trivial elaborations of these two “motifs”. By “nearly trivial elaborations” I mean that their cases can all be shown equivalent to starting with Case 1 or 2 and then adding dynamics that can affect stability but not RPA, so these additional dynamics are inconsequential for RPA. I’ll work through one case study at the end.

So it is easily shown that the 6 minimal modules in S6.2 (pages 47-50) are not minimal. In particular, S6.2.3 Case C is an elaboration of Case 2, and the rest are either exactly Case 1 (S6.2.1 Case A) or are elaborations of it (the rest). This is easily shown constructively by simply using the IFC Theorem to extract the z variable that is the integrator, and the rest of the states don’t contribute to PA and merely need to preserve stability.

We certainly do agree that “Case 1” and “Case 2” are distinct cases. They are, in fact, the “minimal” versions of an opposer module (employing a single opposer node, or a “trivial” opposing set), and a minimal balancer module. We also agree that our original distinctions among modules, suggesting a total of five different modules rather than two, were unnecessary.

As we will elaborate further in the next section, however, we do not agree that our former “Case C” (now “Case a” in our new Figure 8) is simply an elaboration of Case 2. On the contrary: This is an opposer module (involving a single opposer node, O_1 , noted in yellow), connected *in series* with a balancer module (with a single balancer node, B, noted in blue, and connector node, C, noted in green). We invite the reviewer to verify that if the reaction kinetics at node O_1 do not conform to the special form for “opposer kinetics”, the output node “O” will not exhibit RPA. We also emphasize again that this is not intended to be a “minimal module”: This is a minimal network containing two modules connected together (an opposer module and a balancer module).

We also agree that our former Cases A and B, and also D and E, were all examples of an opposer module (with A and B each using a single opposer node, while D and E each use a two-node opposing set). But we do not agree that the former Case F (now Case f in our new Figure 8) is an elaboration of the reviewer’s Case 1 example. Rather, our new Case f is actually two opposer modules connected together in series – one module uses a two-node opposing set involving O_1 and O_2 , while the other module uses a single opposer node, O_3 . We highlight this important subtly by choosing a related small network for Case e, where

the opposer module involving the two-node opposing set (O_1 and O_2) is now connected in series with a balancer node instead.

This distinction – interpreting Case f as two opposer modules connected together, rather than as an “elaboration” of the reviewer’s case 1 – is not just splitting hairs: it is absolutely crucial to our ability to specify the set of all possible solutions to the RPA problem via the correct identification of what we can truly consider a “basis”, along with a set of rules for combining those basis elements to give the set of all possible RPA networks. The reviewer is right to identify that Case f only requires the opposition mechanism, and for this simple case, the three opposer nodes place zeros into the matrix form of the RPA equation in locations that make it clear by inspection that the matrix is singular. But by contrasting cases **e** and **f** in Figure 8, we readily observe that that O_1 and O_2 make a different contribution to the RPA solution from O_3 . In particular, O_1 and O_2 collaborate to “oppose” the route $I \rightarrow O$ in both cases. But in Case f, O_3 “opposes” the route $I \rightarrow O_1 \rightarrow X_1 \rightarrow O_2 \rightarrow X_2 \rightarrow O$. In Case e, on the other hand there are two other routes in the network, $I \rightarrow O_1 \rightarrow X_1 \rightarrow O_2 \rightarrow C \rightarrow O$ and $I \rightarrow O_1 \rightarrow X_1 \rightarrow O_2 \rightarrow B \rightarrow C \rightarrow O$, which are balanced by the balancer module employing B (noted in blue) and C (noted in green). In both Cases e and f, the opposer nodes O_1 and O_2 play a regulatory role (as feedback integrators) *and also* play a transmissive role (participating in transmission routes from I to O. This subtlety is not readily noted from the matrix form of the RPA equation (or the IFC Det condition used by Yi et al).

We take great pains in our revised submission to provide a clear and lucid explanation as to how *a topological analysis* of the set of terms (**R**) comprising the determinant expansion of the RPA equation yields a very clear notion of RPA basis elements. We carefully explain that this may be achieved via a partition of **R** into “*independently adapting subsets*”, and through a logical and rigorous set of arguments, emphasize that

“... from the observation that the terms of **R** are distributed to independently adapting subsets by route (that is, all instances in **R** of a particular route are to be grouped together into a single such subset), it follows that these subsets are disjoint, and must cover **R**. We have seen, moreover, that two *and only two* mechanisms – which we call *opposition* and *balancing* – are able to generate the independently adapting subsets of **R** in an RPA capable network, and that each such mechanism may be implemented by a rich class of sub-network topologies – opposer modules and balancer modules, respectively. Taken together, these considerations imply that a network can exhibit RPA only if it is decomposable into opposer and/or balancer modules – that is, each route for the transmission of biochemical signal from input to output must be either balanced or (fully) opposed by a single network module.”

Thus, basis sets are created in **R** from “generating mechanisms” – opposition and balancing – and these correspond to specific classes topological arrangements of nodes in the

associated network, which we elaborate fully in the preceding sections of our manuscript. In other words, a basis module is comprised of a collection of routes along with the mechanism that either balances or (fully) opposes those routes.

But the key point which turns these basis sets, and their corresponding basis modules, into the full solution space to the RPA problem is the set of rules that specifies how these modules can coexist in larger networks. As we point out,

“A general RPA network could contain an arbitrary number of such modules – corresponding to its RPA equation being partitioned into (the same) arbitrary number of disjoint *independently adapting subsets* – so the question now remains as to how multiple such network modules may coexist (ie. be connected together) in RPA networks.”

We begin our careful explanations as to these “connectivity rules” with the observation that

–
“...the nature of the two distinct RPA-generating mechanisms, and their topological realizations in self-organizing/self-regulating networks, does place some constraints on how RPA modules may be interconnected to form more complex “multi-modular” networks. These constraints are two-fold: First, we note that the three types of reaction kinetics required to implement RPA – opposer kinetics, balancer kinetics and connector kinetics – are mutually exclusive (SI Section 5). That is, any given node can exhibit *at most one of the three types* of reaction kinetics. Second, any given computational node (opposer, balancer or connector) is constrained in *how* it may be regulated: an opposer node, or a collection of collaborating balancer nodes, each has a *single* independent regulator; and a connector node works with a *single* collection of collaborating balancer nodes.

“From this, we can conclude that the “active” part of each module (nodes residing between the “apex” (node “C” in Figure 1a,b and node “D” in Figure 1c), and the “base” (“D” in Figure 1a,b and “C” in Figure 1c) must be distinct from the active part of any other module. A node that plays the role of an opposer in one module, for instance, cannot also be required to operate as a balancer (or a connector) for some other module. Moreover, the requirement for a single independent regulator implies that an opposer node can only perform its computational function for *a single* opposer module. Likewise, a set of collaborating balancer nodes, together with their connector node, delineates a *single* balancer module.

“The requirement for distinctness of the active parts of RPA modules implies that the modules may either be connected “*in parallel*”, or “*in series*” according to the following definitions:

“Two RPA modules are said to be connected **in parallel** if none of the computational nodes within either module participate in route(s) that are opposed/balanced by the other. The

respective route collections for the two modules must therefore diverge upstream of the active parts of the modules, and then reconnect again downstream of the active parts. Informally speaking, parallel modules are connected “side-by-side” within the global topology of the RPA network. When *an opposer module* is connected in parallel with all other RPA modules that comprise the network, for instance, its opposer node(s) do not participate in *any* route of the network; they participate *in feedback loops only*. This is a comparatively straightforward intermodular arrangement, then, for which we present an example in SI Section 6.1.4 for two opposer modules connected in parallel.

“A *parallel* arrangement of modules may be contrasted with the possibility that in some particular RPA module, one (or more) of its computational nodes may *also* participate in some network route *that is not opposed or balanced by the module in question*. For example, an opposer node – which operates within a *feedback* arrangement relative to the route(s) it opposes – may *also* participate in *some route* within the network. Likewise, a balancer node – embedded into the route segments defining its balancer module – may *also* participate in *some other route* in the network (that is, a route that is *not balanced* by the balancer module in question). In either case, the “extramodular route(s)” in which the computational node(s) participate must be either balanced or fully opposed by one (or more) additional RPA module(s) *connected in series with the original module*. Informally speaking, series modules are connected in an upstream-downstream arrangement, since computational nodes for the upstream module “feed into” the downstream module.

“In order to make the series connection of RPA modules precise from a topological perspective, we first recall from the preceding sections that *within* the active part of each module, some nodes exhibit the RPA property, while others do not. Opposer nodes, along with any associated dependent nodes, *do not* exhibit the RPA property. The single independent regulator for an opposer, along with any associated dependent nodes, *do* exhibit the RPA property. Likewise, balancer nodes, along with their single independent regulator (the associated D-node) *do not* exhibit the RPA property, while connector nodes *do* exhibit the RPA property. From these considerations, we can consider any *outgoing* regulations from the active parts of an RPA module - leading ultimately to the network’s output node - to be either “blind regulations” if they come from node(s) that exhibit the RPA property or “live regulations” if they come from node(s) which do not exhibit the RPA property.

“We illustrate the essential principles of series interconnections of modules, which are required in any RPA network containing a module with *live* outgoing regulations, in Figure 5. As shown in Figure 5(a,b), outgoing regulations from an opposer node (or associated dependent nodes) place that opposer *in a route* which must then be either balanced or (fully) opposed by some *other* RPA module (as indicated by the symbol “A” which indicates the position of the required ancillary module). Likewise, in a balancer module, outgoing

regulations from balancer nodes place these nodes in routes that are not balanced by the module; these routes must be either balanced or (fully) opposed by some other ancillary module, as indicated by the symbol “A” in Figure 5c. In either case, any outgoing *blind regulations* generate no requirements for any ancillary modules. Thus, any module with *only blind outgoing regulations* may exist *alone* in an RPA network, and any subnetwork structures downstream of such regulations may be considered part of the module itself. In addition, blind outgoing regulations may “feed into” any other RPA module(s) without affecting the ability of those modules to contribute to RPA in the network as a whole.

Now, the reviewer continues his/her argument that all our previously suggested “minimal modules” other than “Case C” are simply elaborations of the “Case 1” illustration via a constructive method that we have reviewed carefully. In particular -

To concretely illustrate this, next I’ll show how S6.2.4 Case D on page 49 is a trivial elaboration of Case 1 above and is not a “distinct class of network module” from Case 1. The same argument applies to 4 of the 5 classes, with the remaining class being equivalent to Class 2 above.

The nonlinear version of Case 2 has an integrator in its linearization but as a nonlinear system is not equivalent to Case 1, so Cases 1 and 2 are legitimately 2 distinct “motifs” that can yield RPA, but the remaining 3 “classes” are not distinct in any meaningful sense. But to make this clear we will carefully work through S6.2.4 Case D and show how it almost trivially reduces to Case 1 above.

Showing S6.2.4 Case D is a version of Case 1:

Recap Case 1 and add a simple notation for the motif, which we’ll call the Minimal IFC motif:

(GRAPHIC)

Now consider S6.2.4 Case D which has RPA and can be mostly simply written and drawn this way

(GRAPHIC)

We can check the conditions of the IFC Theorem:

$$\begin{aligned} \text{A stable } \det[A \ b; \ c \ d] = 0 &\Rightarrow \\ \text{Integral feedback: } z = [0 \ 1 \ 0 \ -1]x, z' = y \end{aligned}$$

And then change coordinates so that z is the new 2nd state, and then redraw the motifs.

(GRAPHIC)

What we see here is that after a simple change of coordinates, S6.2.4 Case D is a special case of a more general motif that is a simple elaboration of the Minimal IFC. The truly general case is this, where the det = 0 condition is trivially seen by inspection of the matrices:

(GRAPHIC)

My point is that this is a huge class of networks all of which have RPA and are an obvious elaboration of the Minimal IFC. By “elaboration I mean adding dynamics that can affect stability but not RPA (so must preserve stability), but these additional dynamics are inconsequential for RPA. The other cases in Section 6 are more complicated but similar. For example, this motif is an obvious elaboration of the Minimal IFC (see also 6.2.2 Case B), but has a different output.

(GRAPHIC)

We acknowledge the reviewer's constructive argument for obtaining our previous "Case D" from his/her "Case 1" (which was based on our previous "Case A"). We acknowledge the correctness of the reviewer's analysis which uses a coordinate transformation to demonstrate the essential equivalence of the Case D and Case 1 (or equivalently, Case A). Along similar lines, we have now shown that single opposer nodes vacuously satisfy the conditions of Theorem 3 (on opposing sets), thereby demonstrating that an opposer module with a single opposer node is a simple special case of an opposer module with an opposing set (ie. a single opposer is a "trivial" opposing set.)

The reviewer then concludes his/her review with the following statement -

Note that 4 of the 5 "distinct classes" are just such elaborations, with added dynamics that have no effect on RPA, which only depends on the existence of a single integrator state (which may be a linear combination of states in the original coordinates). This does not rule out that there may be more "distinct classes" to be found (the elaboration of the IFF case 2 is less obvious), but the ones in the paper do not qualify.

Note also that the direct use of the IFC theorem vastly simplifies the arguments here, certainly compared with the complexity of those in the supplement. Even if corrected, the results simply don't seem to need this level of complexity. RPA and the IFC theorem is arguably the simplest nontrivial result in control theory, and deserves more attention, particularly in biology, but not greater complexity unless that complexity buys additional insights.

While we agree that the IFC theorem is able to confirm that all our sample networks are indeed capable of exhibiting RPA, and more importantly, can be used to construct even more networks using our examples as starting points, we respectfully point out that the IFC Theorem – as an essentially algebraic statement on RPA-capable networks – does not readily lend itself to identifying **the set of all networks capable of exhibiting RPA** (as we define the term RPA here).

Our revised manuscript extensively and thoroughly addresses this reviewer's comments, clearly showing how we have taken the study of the RPA problem to a whole new level through the identification of network elements that can truly be considered **basis elements**, along with a **general way to combine** those elements, so as to **span the complete solution space** to the RPA problem.

The definitive conclusions we draw in this work provide a new level of understanding of the essential structures of complex networks, and as such, require a completely rigorous mathematical justification. Our fairly extensive Supplementary Information document, which contains most of the deeper technical details of this work, studies the problem painstakingly and rigorously, using a topological framework that is entirely different from the reviewer's arguments, although based on a common underlying foundation (namely, the RPA equation, which is a convenient special case of the criterion given by Yi et al, and referred to repeatedly by the reviewer). We have taken great pains to rewrite this supplement to make it more clear and concise, but acknowledge that this document inevitably entails significant "complexity" (as the reviewer suggests). This Supplement will

inevitably not be of interest to all readers, but the careful and systematic analysis contained therein, with its emphasis on rigor and abstraction (yet always supported with concrete illustrative examples) is crucial to being able to draw the extremely strong conclusion we present: namely, that all networks capable of exhibiting RPA are decomposable into some combination of the two classes of modules we carefully define. In other words, no RPA-capable network can exist that is not constructible from these well-defined topological basis modules.

The pervasive explanatory power of such a strong statement as to the essential structures unifying **all** RPA networks permits us to strongly address the reviewer's statement that "*This does not rule out that there may be more "distinct classes" to be found (the elaboration of the IFF case 2 is less obvious), but the ones in the paper do not qualify.*" In this work, we reveal two things that have never before been shown, or even suggested, to our knowledge: (a) we show conclusively that opposition and balancing are the *only* two RPA mechanisms, and (b) we show, topologically speaking, how these two mechanisms can be *combined, in general*, in the unfathomably complex networks that exist in nature.

Thus, our overall purpose in the revised manuscript is to move beyond the classic small networks comprising just a few nodes (eg. in the Yi et al study, where the authors discuss the small collection of interactions governing RPA in bacterial chemotaxis). These simple cases say nothing about most networks of interest to biologists (or clinicians, even: How might we understand the immense signaling networks underpinning cancer signal transduction, for instance?)

In closing, we would like to thank the reviewer again for such generous support of our work, and for the huge investment of time involved in considering our work so carefully. The reviewer's detailed comments and analysis have tremendously enriched our understanding of the control engineering principles relevant to this work which, we feel, has allowed us to produce a more scholarly article. We cannot thank the reviewer enough for giving us a fresh perspective on how to communicate the significance and distinctiveness of our approach more clearly, and to make our work more accessible to an even wider scientific readership.

REVIEWERS' COMMENTS:

Reviewer #1 (Remarks to the Author):

The authors revised the manuscript well. I have no further questions.

Reviewer #2 (Remarks to the Author):

Overview: For cells to function and survive in a complex, dynamically varying environment, it is imperative for them to contain network modules that display *Perfect Adaptation* (PA) w.r.t. some input signals. PA is the ability of a system to “reject” disturbances in input stimulus by returning to its reference level or “set-point”. If this property PA holds without the need for tuning system parameters, then it is called *robust* PA or RPA. The aim of this paper is to demonstrate that RPA-capable networks have a special structure, that is expressible using a topological basis containing two simple types of network motifs. The key result is that any RPA-capable network, irrespective of its size, can be decomposed into smaller network modules, each of which belongs to these two distinct network classes that form the topological basis.

Recommendation: The results in this paper are interesting and present a important advance in our understanding of RPA in biological reaction networks. In my opinion, the authors have successfully addressed the comments of Reviewer 2 and improved their manuscript substantially in the process. I recommend this paper for publication but the authors must address the issues mentioned below.

1. The RPA conditions (1)-(2) are not new and the authors acknowledge it and mention that these conditions can be derived from the *Internal Model Principle* in control theory. However it seems that the authors are not aware that these conditions have appeared before in the context of RPA for biological networks in [?]. This paper calls it the “cofactor” condition which is equivalent to the determinant conditions given in the paper. The authors must cite this paper and compare its results with their results.
2. A necessary requirement for RPA is global stability, to ensure that there is a unique equilibrium point regardless of the initial condition. However the main paper completely sidesteps the stability issue and it is covered only very briefly in Section 9 of the Supplementary Information document. Even though negative feedback loops are “stability promoting” as mentioned in the Supplementary, they also tend to induce oscillations and possibly even chaotic behavior. The authors must discuss this point in the main paper so as to apprise the readers to some of the limitations of the “algebraic” approach presented in the paper, as opposed to the more direct “dynamical systems” approach found in other papers.
3. This is just a minor point. On lines 339-341 on page 12 it says that “The balancing mechanism thus requires a computational collaboration between two

distinct types of nodes: a collection of one or more balancer nodes, along with a single connector node.”
Shouldn't there be at least two balancer nodes for this mechanism to work?

References

- [1] Z. F. Tang and D. R. McMillen. Design principles for the analysis and construction of robustly homeostatic biological networks. *Journal of theoretical biology*, 408:274–289, 2016.

Response to Reviewers

We sincerely thank all three reviewers for such a careful and detailed review of our work, and for the abundance of helpful and insightful feedback that has given us the opportunity to make such substantial improvements to our original submission.

We reproduce each remaining reviewer's report below, *italicized and indented* for clarity, and follow each individual point with our response and explanations as to how we have changed our manuscript.

Reviewer #1

The authors revised the manuscript well. I have no further questions.

Reviewer #3 (for previous Reviewer #2)

Overview: *For cells to function and survive in a complex, dynamically varying environment, it is imperative for them to contain network modules that display Perfect Adaptation (PA) w.r.t. some input signals. PA is the ability of a system to “reject” disturbances in input stimulus by returning to its reference level or “set-point”. If this property PA holds without the need for tuning system parameters, then it is called robust PA or RPA. The aim of this paper is to demonstrate that RPA-capable networks have a special structure, that is expressible using a topological basis containing two simple types of network motifs. The key result is that any RPA-capable network, irrespective of its size, can be decomposed into smaller network modules, each of which belongs to these two distinct network classes that form the topological basis.*

Recommendation: *The results in this paper are interesting and present an important advance in our understanding of RPA in biological reaction networks. In my opinion, the authors have successfully addressed the comments of Reviewer 2 and improved their manuscript substantially in the process. I recommend this paper for publication but the authors must address the issues mentioned below.*

We thank this reviewer very warmly for such a thorough review of our work, and for his/her supportive feedback and suggestions for some additional improvements to our final manuscript. We have considered each point very carefully, and have made all the suggested revisions as detailed below.

1. The RPA conditions (1)-(2) are not new and the authors acknowledge it and mention that these conditions can be derived from the Internal Model Principle in control theory. However it seems that the authors are not aware that these conditions have appeared before in the context of RPA for biological networks in [1]. This paper calls it the “cofactor” condition which is equivalent to the determinant conditions given in the paper. The authors must cite this paper and compare its results with their results.

We thank the reviewer very much for drawing our attention to this key recent paper, and for giving us the opportunity to cite it, and review its findings in the context of our new study. We are very happy to be able to update our paper with a more complete exposition of all prior relevant work. Under the heading “Methods Overview and Relationship to the IMP” within our newly-created Methods section, we now include the following text:

We note that an alternative, but mathematically equivalent, version of the RPA equation has also been developed in the recent work of Tang and McMillen [1]. Those authors refer to the condition as “the cofactor condition”, and apply this approach to the issue of designing novel homeostatic systems. In particular, their design algorithm has been used to generate topologies and parameter constraints that “will support homeostatic behavior for a given set of network components and a desired set of general regulatory constraints to be applied between them” [1].

2. A necessary requirement for RPA is global stability, to ensure that there is a unique equilibrium point regardless of the initial condition. However the main paper completely sidesteps the stability issue and it is covered only very briefly in Section 9 of the Supplementary Information document. Even though negative feedback loops are “stability promoting” as mentioned in the Supplementary, they also tend to induce oscillations and possibly even chaotic behavior. The authors must discuss this point in the main paper so as to apprise the readers to some of the limitations of the “algebraic”

approach presented in the paper, as opposed to the more direct “dynamical systems” approach found in other papers.

We thank the reviewer for this suggestion, and now provide a clarification on this point in the main paper. In the section “Additional Notes on Methods” in our newly-created Methods section, we now insert the following text:

For completeness, we also observe that although the topological structures we identify here are necessary conditions for solving the RPA problem in complete generality, these conditions are not sufficient by themselves to guarantee the implementation of RPA across all possible parameter regimes. In practice, RPA also requires global stability to ensure that there is a unique and stable steady-state regardless of initial conditions. We discuss stability issues briefly in Supplementary Note 8, where we point out that feedback loops, if present at all, should be negative feedback loops since these are stability promoting. We nevertheless acknowledge that negative feedback could potentially induce oscillations or even chaotic behavior. More direct dynamical systems approaches are required to examine these possibilities for specific RPA topologies and specific parameter regimes.

3. This is just a minor point. On lines 339-341 on page 12 it says that “The balancing mechanism thus requires a computational collaboration between two distinct types of nodes: a collection of one or more balancer nodes, along with a single connector node.” Shouldn’t there be at least two balancer nodes for this mechanism to work?

We thank the reviewer very much for paying such close attention to the details of our work. This is an important question. We provide several specific examples in the manuscript in which one balancer node is sufficient. (Figure 2b, Figure 8b and Figure 8c, for example. In each of these cases, the single balancer node is indicated in blue).

References

[1] Z. F. Tang and D. R. McMillen. *Design principles for the analysis and construction of*

robustly homeostatic biological networks. Journal of theoretical biology, 408:274–289, 2016.